# Humidity-driven ABA depletion determines plant-pathogen competition for leaf water

Shigetaka Yasuda [1] ✉, Akihisa Shinozawa[2,8], Yuanjie Weng[3,8], Arullthevan Rajendram[1,8], Taishi Hirase[1,8], Haruka Ishizaki[1], Ryuji Suzuki[4], Shioriko Ueda[1], Rahul Sk [5], Yumiko Takebayashi [3], Izumi Yotsui[2], Masatsugu Toyota [4,6,7], Masanori Okamoto [3] & Yusuke Saijo [1] ✉

Bacterial phytopathogens, such as *Pseudomonas syringae* pv. tomato (*Pst*) DC3000, induce water-soaked lesions in the leaf apoplast under high humidity, facilitating infection. However, it remains largely unclear how plants regulate their resistance to restrict bacterial infection in response to humidity. Here, we demonstrate that abscisic acid (ABA)-catabolizing ABA 8'-hydroxylase, encoded by *CYP707A3*, plays a critical role in this resistance in *Arabidopsis thaliana*. Elevated humidity induces *CYP707A3* expression, which is essential for reducing ABA levels and promoting stomatal opening, thereby limiting bacterial water-soaking and infection following leaf invasion. High humidity also increases cytosolic $Ca^{2+}$ levels via the $Ca^{2+}$ channels CNGC2 and CNGC4, with partial involvement from CNGC9, activating the calmodulin-binding transcription activator CAMTA3 to drive *CYP707A3* induction. However, *Pst* DC3000 counteracts this defense response using type III secretion effectors, including AvrPtoB, facilitating water-soaking. These findings provide insights into the mechanisms underlying the competition between plants and bacteria for leaf water under elevated humidity.

Bacterial phytopathogens cause foliar diseases that threaten plant survival and crop yields by proliferating within the leaf apoplast[1]. An important layer of defense against bacterial infection is provided by cell surface receptor-like kinases/proteins (RLKs/RLPs), collectively known as pattern recognition receptors (PRRs). PRRs detect microbe-associated molecular patterns (MAMPs) and endogenous damage-associated molecular patterns (DAMPs) to activate pattern-triggered immunity (PTI)[2–4]. Following ligand perception, PRRs form active complexes with co-receptors to trigger intracellular signaling cascades. This involves rapid cytosolic $Ca^{2+}$ influx, a burst of reactive oxygen species (ROS), activation of mitogen-activated protein kinase (MAPK) cascades, production of defense-related phytohormones such

as salicylic acid (SA), stomatal closure, transcriptional reprogramming, and production of antibacterial compounds[2–4]. To overcome PTI, bacterial pathogens deliver effector proteins into host tissues and cells via the type III secretion (T3S) system[1]. The bacterial phytopathogen *Pseudomonas syringae* pv. *tomato* (*Pst*) DC3000 has 36 T3S effectors[5]. Recent studies have shown that *Pst* DC3000 utilizes specific T3S effectors to manipulate the leaf apoplast environment, establishing conditions favorable for infection[6].

High humidity promotes foliar diseases caused by bacterial pathogens through multiple mechanisms. Initially, impaired stomatal closure facilitates bacterial invasion into leaves[7]. Once inside, bacterial pathogens induce apoplast hydration under high humidity, a critical

¹Graduate School of Science and Technology, Nara Institute of Science and Technology, Ikoma, Japan. ²Department of Bioscience, Tokyo University of Agriculture, Tokyo, Japan. ³Center for Sustainable Resource Science, RIKEN, Yokohama, Japan. ⁴Department of Biochemistry and Molecular Biology, Saitama University, Saitama, Japan. ⁵The NODAI Genome Research Center (NGRC), Tokyo University of Agriculture, Tokyo, Japan. ⁶Suntory Rising Stars Encouragement Program in Life Sciences (SunRiSE), Suntory Foundation for Life Sciences, Kyoto, Japan. ⁷College of Plant Science and Technology, Huazhong Agricultural University, Wuhan, China. ⁸These authors contributed equally: Akihisa Shinozawa, Yuanjie Weng, Arullthevan Rajendram, Taishi Hirase. ✉e-mail: shige-yasuda@bs.naist.jp; saijo@bs.naist.jp

process for disease development known as water-soaking[8]. To induce water-soaking, *Pst* DC3000 employs the highly conserved T3S effectors HopM1 and AvrE, which manipulate the host plant's abscisic acid (ABA) pathways, a phytohormone critical for water retention[9,10]. Plants typically increase ABA levels under drought conditions to promote stomatal closure and conserve water[11]. However, under high humidity, when plants naturally reduce ABA levels, *Pst* DC3000 stimulates ABA biosynthesis and signaling in the host through HopM1 and AvrE, thereby enhancing water-soaking and promoting disease progression[9,10]. Similarly, *Xanthomonas* bacteria exploit host ABA pathways through their T3S effectors to facilitate leaf infection in rice, wheat, and *Arabidopsis thaliana* (hereafter Arabidopsis)[12–14]. Despite recent advances in the identification and characterization of bacterial effectors involved in water-soaking, our understanding of the plant resistance mechanisms that counteract bacterial water acquisition remains limited[15,16].

In nature, high humidity often accompanies prolonged rainfall, inducing a wide range of physiological effects in plants, such as defects in cuticle formation, stomatal opening, changes in water uptake and transpiration, and leaf petiole movement (hyponasty)[7,17–19]. In Arabidopsis, high humidity rapidly induces transcriptional and post-transcriptional changes in leaves, modulating phytohormone pathways[17,19]. ABA levels are tightly regulated through the coordination of its biosynthesis and catabolism[20]. Under high humidity, ABA levels decrease via the action of *CYP707A1* and *CYP707A3*, which encode ABA 8′-hydroxylases, thereby facilitating stomatal opening[17]. Concurrently, the production and signaling of defense-promoting salicylic acid (SA) are suppressed, partly due to impaired ubiquitination of NON-EXPRESSOR OF PATHOGENESIS-RELATED GENES 1, an SA receptor and key transcriptional co-activator of SA-responsive genes, reducing its binding to target gene promoters[21]. Additionally, high humidity stimulates the biosynthesis of the gaseous phytohormone ethylene, which promotes leaf hyponasty[19]. Despite these extensive changes in phytohormone production and signaling, the mechanisms by which the modulation of host phytohormone pathways under high humidity impacts plant–pathogen interactions remain poorly understood.

In this study, we demonstrate that the high humidity-induced *CYP707A3* plays a pivotal role in restricting bacterial water-soaking by promoting stomatal opening in Arabidopsis leaves. While this work was in preparation, Hussain et al. reported that elevated humidity increases cytosolic $Ca^{2+}$ concentrations in a CYCLIC NUCLEOTIDE-GATED CHANNEL 2 (CNGC2)- and CNGC4-dependent manner, thereby inducing *CYP707A3* expression through CALMODULIN-BINDING TRANSCRIPTION ACTIVATOR 3 (CAMTA3)[22]. In parallel, our investigation independently identified this signaling module as a key determinant of humidity-triggered *CYP707A3* induction. Moreover, we demonstrate that CNGC9 also contributes to this response, and that CAMTA3 is required for resistance to bacterial water-soaking. However, during infection, *Pst* DC3000 counteracts this signaling pathway and suppresses high humidity-induced transcriptional changes through its T3S effectors, including AvrPtoB. These findings provide valuable insights into the mechanisms regulating plant–pathogen competition for leaf water under high humidity.

## Results

### *CYP707A3* is required for resistance to bacterial water-soaking under high humidity

Upon exposure to high humidity, *CYP707A1* and *CYP707A3*, but not *CYP707A2* or *CYP707A4*, were transiently induced in wild-type (WT) leaves within 0.5 h (Fig. 1a). To investigate the role of *CYP707A1* and *CYP707A3* in bacterial resistance, we measured ABA levels and assessed bacterial water-soaking in the corresponding mutants (Supplementary Fig. 1a, b). In WT plants, high humidity triggered a marked reduction in ABA levels within 2 h (Fig. 1b). This ABA decline was completely abolished in the *cyp707a1 cyp707a3* double mutant, which instead exhibited constitutively high ABA accumulation (Fig. 1b), indicating that these genes are required for humidity-induced ABA catabolism.

We next assessed water-soaking in *cyp707a* mutants (Supplementary Fig. 1a, b) using *Pst* Δ*hrcC*, a strain lacking a functional T3S system. Resistance to water-soaking was evaluated by monitoring leaf water retention following bacterial infiltration under high humidity.

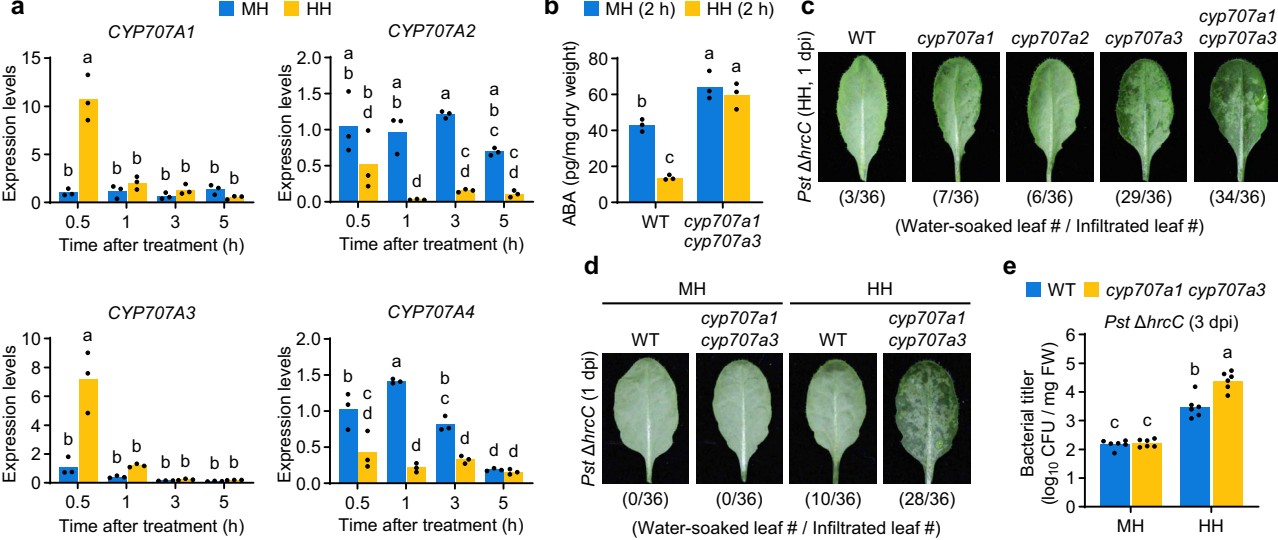

**Fig. 1 | *CYP707A3* is required for resistance to bacterial water-soaking under high humidity. a** Expression levels of *CYP707A* genes in humidity-treated leaves. Wild-type (WT) plants were exposed to moderate humidity (MH) or high humidity (HH) for the indicated time. Bars represent means (*n* = 3). **b** ABA levels in humidity-treated leaves. The indicated plants were exposed to MH or HH for 2 h. Bars represent means (*n* = 3). **c, d** Water-soaking in *Pst* Δ*hrcC*-inoculated leaves. The indicated plants were infiltrated with *Pst* Δ*hrcC* (OD$_{600}$ = 0.02) and maintained

under MH or HH for 1 day. The number of water-soaked leaves was pooled from two independent experiments. **e** Bacterial growth in *Pst* Δ*hrcC*-inoculated leaves. The indicated plants were infiltrated with *Pst* Δ*hrcC* (OD$_{600}$ = 0.02) and maintained under MH or HH for 3 days. Bars represent means (*n* = 6). Sample size (*n*) indicates biological replicates. Different letters indicate statistically significant differences (*P* < 0.05; two-way ANOVA followed by Tukey's test). dpi days post-infiltration. Experiments in (**a, e**) were repeated twice with similar results.

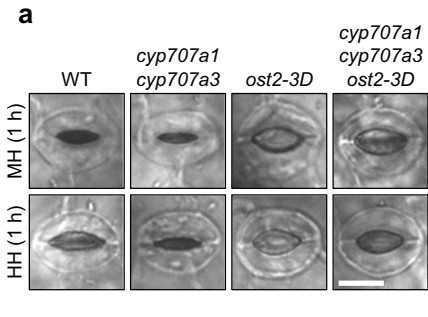

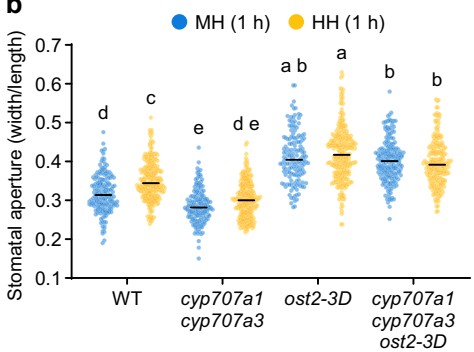

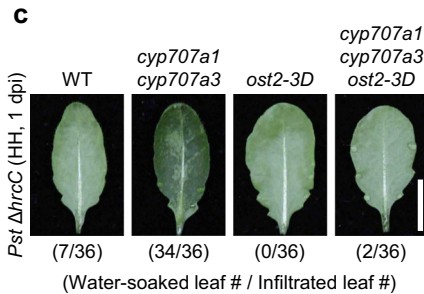

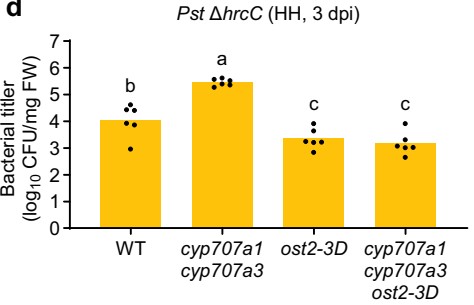

**Fig. 2 | *CYP707A3*-mediated water-soaking resistance depends on stomatal opening. a**, **b** Representative images of stomata (**a**) and stomatal aperture (**b**) in humidity-treated leaves. The indicated plants were exposed to moderate humidity (MH) or high humidity (HH) for 1 h. Bars represent the median (WT MH, $n = 172$; WT HH, $n = 246$; *cyp707a1 cyp707a3* MH, $n = 153$; *cyp707a1 cyp707a3* HH, $n = 236$; *ost2-3D* MH, $n = 149$; *ost2-3D* HH, $n = 264$; *cyp707a1 cyp707a3 ost2-3D* MH, $n = 189$; *cyp707a1 cyp707a3 ost2-3D* HH, $n = 219$). Scale bar = 10 μm. **c** Water-soaking in *Pst ΔhrcC*-inoculated leaves. The indicated plants were infiltrated with *Pst ΔhrcC* ($OD_{600} = 0.02$) and maintained under HH for 1 day. The number of water-soaked leaves was pooled from two independent experiments. Scale bar = 1 cm. **d** Bacterial growth in *Pst ΔhrcC*-inoculated leaves. The indicated plants were infiltrated with *Pst ΔhrcC* ($OD_{600} = 0.02$) and maintained under HH for 3 days. Bars represent means ($n = 6$). Sample size ($n$) in (**b**, **d**) indicate stomata and biological replicates, respectively. Different letters indicate statistically significant differences ($P < 0.05$; two-way ANOVA followed by Tukey's test in (**b**); one-way ANOVA followed by Tukey's test in (**d**)). WT wild-type, dpi days post-infiltration. Experiments in (**b**, **d**) were repeated twice with similar results.

At 1 day post-infiltration (dpi), pronounced water-soaking was observed in *cyp707a3* and *cyp707a1 cyp707a3*, but not in WT, *cyp707a1*, or *cyp707a2* (Fig. 1c). The phenotype was more severe in *cyp707a1 cyp707a3* than in *cyp707a3*, suggesting a predominant role for *CYP707A3* with a partial contribution from *CYP707A1* in water-soaking resistance. Accordingly, we used *cyp707a1 cyp707a3* for most subsequent analyses. Under moderate humidity, however, even *cyp707a1 cyp707a3* displayed no detectable water-soaking (Fig. 1d). Consistently, *Pst ΔhrcC* growth was significantly higher in *cyp707a1 cyp707a3* than in WT under high humidity, but was indistinguishable under moderate humidity (Fig. 1e). These results demonstrate that *CYP707A3* is crucial for restricting water-soaking during *Pst* DC3000 infection, thereby limiting bacterial proliferation specifically under high humidity.

### *CYP707A3*-mediated water-soaking resistance depends on stomatal opening

*Pst* DC3000 induces stomatal closure following leaf invasion to facilitate water-soaking[9,10]. Consistent with previous reports[17], loss of *CYP707A3* and *CYP707A1* impaired high-humidity-induced stomatal opening (Fig. 2a, b). To test whether *CYP707A3*-mediated bacterial resistance is linked to stomatal regulation, we introduced the *ost2-3D* mutation, an active allele of the plasma membrane H⁺-ATPase AHA1 that drives constitutive stomatal opening[23], into the *cyp707a1 cyp707a3* background (Supplementary Fig. 2a, b). As expected, the *cyp707a1 cyp707a3 ost2-3D* triple mutant exhibited enhanced stomatal opening compared with *cyp707a1 cyp707a3* under both high and moderate humidity (Fig. 2a, b). Correspondingly, water-soaking was almost completely abolished in *cyp707a1 cyp707a3 ost2-3D* following

*Pst ΔhrcC* infiltration, similar to *ost2-3D* (Fig. 2c). This suppression was accompanied by markedly reduced *Pst ΔhrcC* growth in both *cyp707a1 cyp707a3 ost2-3D* and *ost2-3D* (Fig. 2d). These findings indicate that *CYP707A3* confers resistance to water-soaking, at least in part, by promoting stomatal opening under high humidity.

### ABA depletion enhances bacterial resistance by restricting water-soaking independently of SA under high humidity

SA antagonizes many ABA-dependent processes and has been reported to suppress water-soaking under light conditions[16,24,25]. However, both SA and ABA are required for stomatal closure during early defense responses to bacterial phytopathogens[26]. To assess whether SA influences *CYP707A3*-mediated water-soaking resistance, we introduced the *sa induction deficient 2-2* (*sid2-2*) mutation, which disrupts ISOCHORISMATE SYNTHASE 1 (ICS1)-dependent SA biosynthesis[27], into the *cyp707a1 cyp707a3* background (Supplementary Fig. 3a, b). The *cyp707a1 cyp707a3 sid2-2* triple mutant, but not *sid2-2*, exhibited extensive water-soaking following *Pst ΔhrcC* infiltration (Supplementary Fig. 3c), indicating that ICS1-dependent SA biosynthesis is dispensable for *CYP707A3*-mediated water-soaking resistance.

We next tested whether ABA depletion alone is sufficient to confer bacterial resistance under high humidity in an SA-independent manner. To this end, we examined water-soaking and bacterial growth following *Pst* DC3000 infiltration in the *abscisic aldehyde oxidase 3* (*aao3*) mutant, which is deficient in ABA biosynthesis[28], with or without the *sid2-2* mutation (Supplementary Fig. 3d, e). Consistent with previous studies on ABA dependence[9,10], *Pst* DC3000-induced water-soaking was markedly reduced in *aao3*, irrespective of *sid2-2* mutation (Supplementary Fig. 3f). This reduction correlated with decreased *Pst*

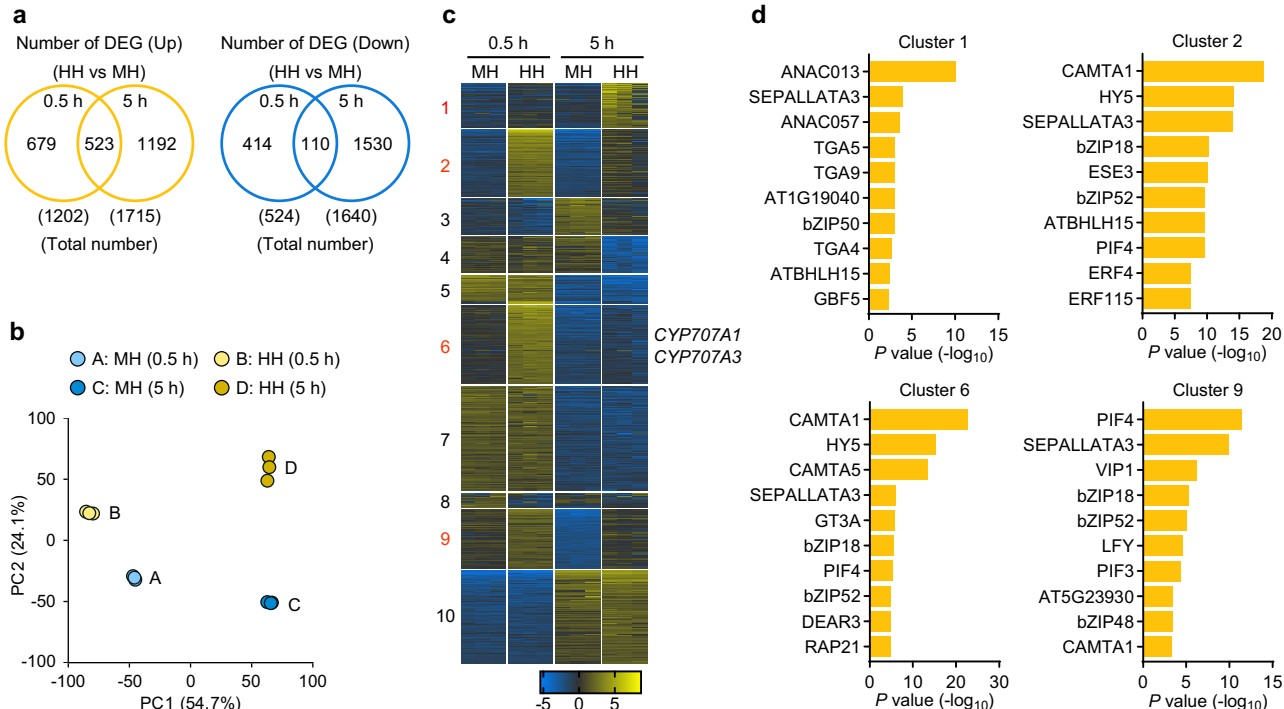

**Fig. 3 | CAMTA-binding motifs are overrepresented in early-phase high humidity-induced genes. a–c** Transcriptome profiling of humidity-treated leaves by RNA-seq. Wild-type plants were exposed to moderate humidity (MH) or high humidity (HH) for the indicated time. **a** Venn diagrams showing differentially expressed genes (DEGs) (fold change > 2 and FDR < 0.05) at 0.5 and 5 h after treatment. **b** Principal component analysis (PCA) of transcriptomic data. **c** K-means clustering of the 1000 most variable genes. **d** Top 10 enriched *cis*-regulatory elements among high humidity-induced genes identified in the clusters shown in (**c**). Enrichment significance was assessed using a two-sided hypergeometric test, and *P* values were adjusted for multiple comparisons using the Benjamini-Hochberg FDR method.

DC3000 growth at 3 dpi in both *aao3* and *aao3 sid2*, although the effect was less pronounced in the latter (Supplementary Fig. 3g). Under moderate humidity, however, suppression of *Pst* DC3000 growth by the *aao3* mutation was observed only in the presence of functional *SID2* (Supplementary Fig. 3h), as reported previously[25]. Collectively, these results demonstrate that ABA depletion confers resistance to *Pst* DC3000 under high humidity by restricting water-soaking development, independently of the SA-ABA antagonism observed under moderate humidity.

## CYP707A3/A1-mediated ABA depletion and PTI contribute additively to bacterial resistance
High humidity differentially affects PTI responses triggered by flg22, a MAMP derived from bacterial flagellin[21]. To test whether high humidity-triggered ABA depletion via *CYP707A3* and *CYP707A1* contributes to this modulation, we compared PTI marker gene expression in WT and *cyp707a1 cyp707a3* following *Pst ΔhrcC* infiltration under moderate and high humidity. After 24 h acclimation to the respective humidity conditions, we measured *FLG22-INDUCED RECEPTOR-LIKE KINASE 1* (*FRK1*), *NON RACE-SPECIFIC DISEASE RESISTANCE 1/HAIRPIN-INDUCED GENE 1-LIKE 10* (*NHL10*), and *PHOSPHATE-INDUCED 1* (*PHI-1*) expression at 1 h post-infiltration (hpi). Under high humidity, *FRK1* induction was enhanced, *NHL10* induction was unchanged, and *PHI-1* induction was abolished (Supplementary Fig. 4a). These patterns were similar in WT and *cyp707a1 cyp707a3*, suggesting that CYP707A3/A1-mediated ABA depletion has limited impact on the humidity-dependent PTI gene regulation.

We next examined the relationship between *CYP707A3*-mediated water-soaking resistance and PTI by crossing *cyp707a1 cyp707a3* with the *brassinosteroid insensitive 1-associated receptor kinase 1-5* (*bak1-5*) *bak1-like 1* (*bkk1*) *chitin elicitor receptor kinase 1* (*cerk1*) triple mutant (hereafter *bbc*), which disrupts PRR activation mediated by the co-

receptors BAK1, BKK1, and CERK1[29] (Supplementary Fig. 4b, c). The *cyp707a1 cyp707a3 bbc* quintuple mutant exhibited pronounced water-soaking following *Pst ΔhrcC* infiltration, similar to *cyp707a1 cyp707a3* (Supplementary Fig. 4d). By contrast, *bbc* showed WT-like or slightly enhanced resistance to water-soaking (Supplementary Fig. 4d), indicating that humidity-triggered ABA depletion restricts water-soaking independently of BAK1/BKK1/CERK1-dependent PTI branches. Notably, *Pst ΔhrcC* growth was significantly higher in *cyp707a1 cyp707a3 bbc* compared to either *cyp707a1 cyp707a3* or *bbc* (Supplementary Fig. 4e), indicating additive effects. Together, these results indicate that CYP707A3/A1-mediated ABA depletion and BAK1/BKK1/CERK1-dependent PTI pathways contribute additively to *Pst* DC3000 resistance under high humidity.

## Global transcriptome analysis reveals CAMTA-binding motif enrichment in early-phase high humidity-induced genes
To capture transcriptional reprogramming triggered by high humidity, we performed RNA-sequencing (RNA-seq) on WT leaves exposed to moderate or high humidity for 0.5 and 5 h. In total, 2394 upregulated and 2054 downregulated genes were identified as differentially expressed genes (DEGs; fold change > 2 and FDR < 0.05) under high humidity compared with moderate humidity (Fig. 3a and Supplementary Data 1). The DEGs identified at 0.5 h and 5 h showed limited overlap (upregulated, 43.5% of 0.5 h and 30.5% of 5 h; downregulated, 30% of 0.5 h and 6.7% of 5 h) (Fig. 3a), suggesting that distinct regulatory mechanisms are involved during early and late stages of the response. This distinction was further supported by principal component analysis (PCA), which revealed clear separation of transcriptomes according to both humidity conditions and exposure durations (Fig. 3b).

K-means clustering of the 1000 most variable genes yielded 10 distinct clusters (Fig. 3c and Supplementary Data 2). High humidity-

induced genes were distributed across four clusters, with *CYP707A1* and *CYP707A3* notably present in Cluster 6 (Fig. 3c). Clusters 2 and 6 exhibited rapid transcriptional upregulation within 0.5 h of treatment, Cluster 9 peaked at 0.5 h and gradually declined by 5 h while remaining elevated under high humidity relative to moderate humidity, and Cluster 1 was primarily induced at 5 h (Fig. 3c). Gene ontology analysis highlighted overrepresented biological processes in these clusters, including "hypoxia response" (Clusters 1 and 2), "jasmonic acid (JA) response" (Cluster 6), and "cell wall biogenesis" (Cluster 9) (Supplementary Fig. 5). Notably, *Cis*-regulatory motif analysis revealed significant enrichment of CAMTA-binding motifs, particularly in Clusters 2 and 6 (Fig. 3d), implicating CAMTA transcription factors as potential key regulators of early transcriptional responses to high humidity.

### High humidity-induced *CYP707A3* expression requires CAMTA-binding motifs in its promoter

The promoter region of *CYP707A3* contains two tandem CAMTA-binding motifs (CGCG-box), whereas the *CYP707A1* promoter lacks these elements (Fig. 4a). To assess the contribution of these motifs to high humidity-induced *CYP707A3* expression, we transiently expressed a GUS reporter driven by either a native *CYP707A3* promoter (harboring both CGCG boxes) or a modified promoter lacking these motifs (ΔCGCG) in *NahG* plants (Supplementary Fig. 6a), which are depleted of SA to facilitate *Agrobacterium*-mediated gene delivery. *GUS* expression driven by the native promoter was significantly induced 0.5 h after high humidity exposure, whereas the ΔCGCG promoter failed to confer this induction (Fig. 4b). In these plants, high humidity induction of endogenous *CYP707A3* expression remained unaffected (Fig. 4b). These results indicate that CAMTA-binding motifs mediate, at least in part, the high humidity responsiveness of the *CYP707A3* promoter.

To evaluate the physiological relevance of this regulation, we generated *cyp707a3* transgenic plants expressing CYP707A3 fused to a C-terminal 3×FLAG tag (CYP707A3-FLAG), driven by either the native or ΔCGCG promoter (Supplementary Fig. 6b, c). Consistent with the GUS reporter results, high humidity-induced *CYP707A3-FLAG* expression was markedly reduced in the ΔCGCG promoter line compared with the native promoter line (Supplementary Fig. 6d). Following *Pst ΔhrcC* infiltration, the impaired water-soaking resistance of *cyp707a3* was fully restored by *CYP707A3-FLAG* expressed under the native promoter, but only partially restored under the ΔCGCG promoter (Supplementary Fig. 6e). Together, these findings indicate that CAMTA-binding motifs in the *CYP707A3* promoter are critical for its high humidity-induced expression and for conferring resistance to bacterial water-soaking.

### High humidity triggers intracellular $Ca^{2+}$ signaling to induce *CYP707A3* expression

CAMTA transcription factors are activated upon binding to $Ca^{2+}$-calmodulin (CaM)[30]. Given their putative role in high humidity-induced transcription, we hypothesized that elevated humidity triggers an increase in cytosolic $Ca^{2+}$ levels. Using *35S::GCaMP3* plants expressing the $Ca^{2+}$ biosensor GCaMP3, we detected a rapid rise in cytosolic $Ca^{2+}$ in leaf cells within 10 min of high humidity exposure (Supplementary Fig. 7a, b and Supplementary Movie 1). To test whether cytosolic $Ca^{2+}$ influx contributes to high humidity-induced *CYP707A3* and *CYP707A1* expression, we applied the $Ca^{2+}$ channel inhibitors $LaCl_3$ and $GdCl_3$ prior to humidity treatment. Both inhibitors significantly suppressed *CYP707A3* induction but did not suppress *CYP707A1* (Supplementary Fig. 7c), indicating a specific requirement for $Ca^{2+}$ signaling in *CYP707A3* induction. Temporal expression profiling further revealed that high humidity-triggered cytosolic $Ca^{2+}$ elevation precedes *CYP707A3* induction (Supplementary Fig. 7b, d). These findings indicate that humidity elevation

triggers cytosolic $Ca^{2+}$ influx in leaves, which in turn mediates *CYP707A3* expression.

### CNGC2/4/9 play a critical role in high humidity-induced *CYP707A3* expression

CNGCs, comprising 20 members in Arabidopsis, mediate cytosolic $Ca^{2+}$ influx in diverse physiological processes[31]. To assess their role in high humidity-triggered $Ca^{2+}$ signaling, we examined *CYP707A3* induction in loss-of-function mutants of individual *CNGCs*. An initial screen in 2-week-old plants revealed that high humidity-induced *CYP707A3* expression was significantly reduced in *cngc2*, *cngc4*, and *cngc9* (Supplementary Fig. 8a, b). Subsequent analysis in 5-week-old plants showed that high humidity induction of both *CYP707A3* and *CYP707A1* was impaired in *cngc2* and *cngc4*, whereas only *CYP707A3* induction was reduced in *cngc9* (Fig. 4c). These results suggest that CNGC2 and CNGC4 act cooperatively to mediate $Ca^{2+}$ influx required for high humidity-induced *CYP707A3* expression, with partial contribution from CNGC9. Collectively, these findings suggest that CNGC2/4-mediated cytosolic $Ca^{2+}$ elevation, together with partial support from CNGC9, is crucial for *CYP707A3* and *CYP707A1* induction in response to high humidity.

### CAMTA3 drives *CYP707A3* induction, ABA depletion, and resistance to bacterial water-soaking under high humidity

Among the six CAMTA transcription factors in Arabidopsis, CAMTA3 is a central regulator of responses to diverse stimuli[32–35]. To assess its role in high humidity-induced *CYP707A3* expression, we used an inactive CAMTA3-A855V variant, which carries A855V substitution in the IQ domain that disrupts CAMTA3 function without triggering SA defense activation[36]. Introduction of *CAMTA3* or *CAMTA3-A855V* under the control of a native *CAMTA3* promoter in the *camta2 camta3* mutant background did not alter plant growth or basal SA levels (Supplementary Fig. 9a–c). *CAMTA3* expression restored WT-like responses to high humidity, including *CYP707A3* induction, ABA depletion, and stomatal opening (Fig. 4d–f). In contrast, these responses were markedly impaired in *CAMTA3-A855V*-expressing plants (Fig. 4d–f), indicating that CAMTA3 activity is required for their induction. Consistent with the absence of CGCG motifs in the *CYP707A1* promoter, high humidity-induced *CYP707A1* expression was unaffected in either *CAMTA3*- or *CAMTA3-A855V*-expressing plants (Fig. 4d), underscoring distinct regulatory mechanisms for *CYP707A3* and *CYP707A1* expression. Moreover, the levels of SA and jasmonoyl-isoleucine (JA-Ile), the major bioactive form of JA, remained unchanged across plant genotypes and humidity treatments (Supplementary Fig. 9c). Together, these findings indicate that CAMTA3 promotes ABA depletion and stomatal opening under high humidity by inducing *CYP707A3*, without altering SA or JA accumulation.

We next tested whether CAMTA3 activity is required to suppress bacterial water-soaking and proliferation under high humidity. Plants expressing *CAMTA3-A855V*, but not *CAMTA3*, exhibited extensive water-soaking and permitted increased growth of *Pst Δhrc* under high humidity (Fig. 4g, h), whereas bacterial growth was unaffected under moderate humidity (Fig. 4i). These findings demonstrate that CAMTA3 activity is required for resistance to bacterial water-soaking and infection specifically under high humidity. This is consistent with a model in which elevated humidity triggers $Ca^{2+}$ influx through CNGC2/4/9, activating CAMTA3 to drive *CYP707A3*-mediated resistance.

### *Pst* DC3000 suppresses high humidity-induced *CYP707A3* expression via type III secretion effectors

*Pst* DC3000 enhances host ABA responses by manipulating ABA biosynthesis, signaling, and translocation to promote water-soaking under high humidity[9,10]. We tested whether *Pst* DC3000 also targets high humidity-triggered ABA depletion via *CYP707A3* and *CYP707A1*. WT leaves were inoculated with *Pst* DC3000, *Pst* Δ*hrcC*, and *Pst* Δ28E

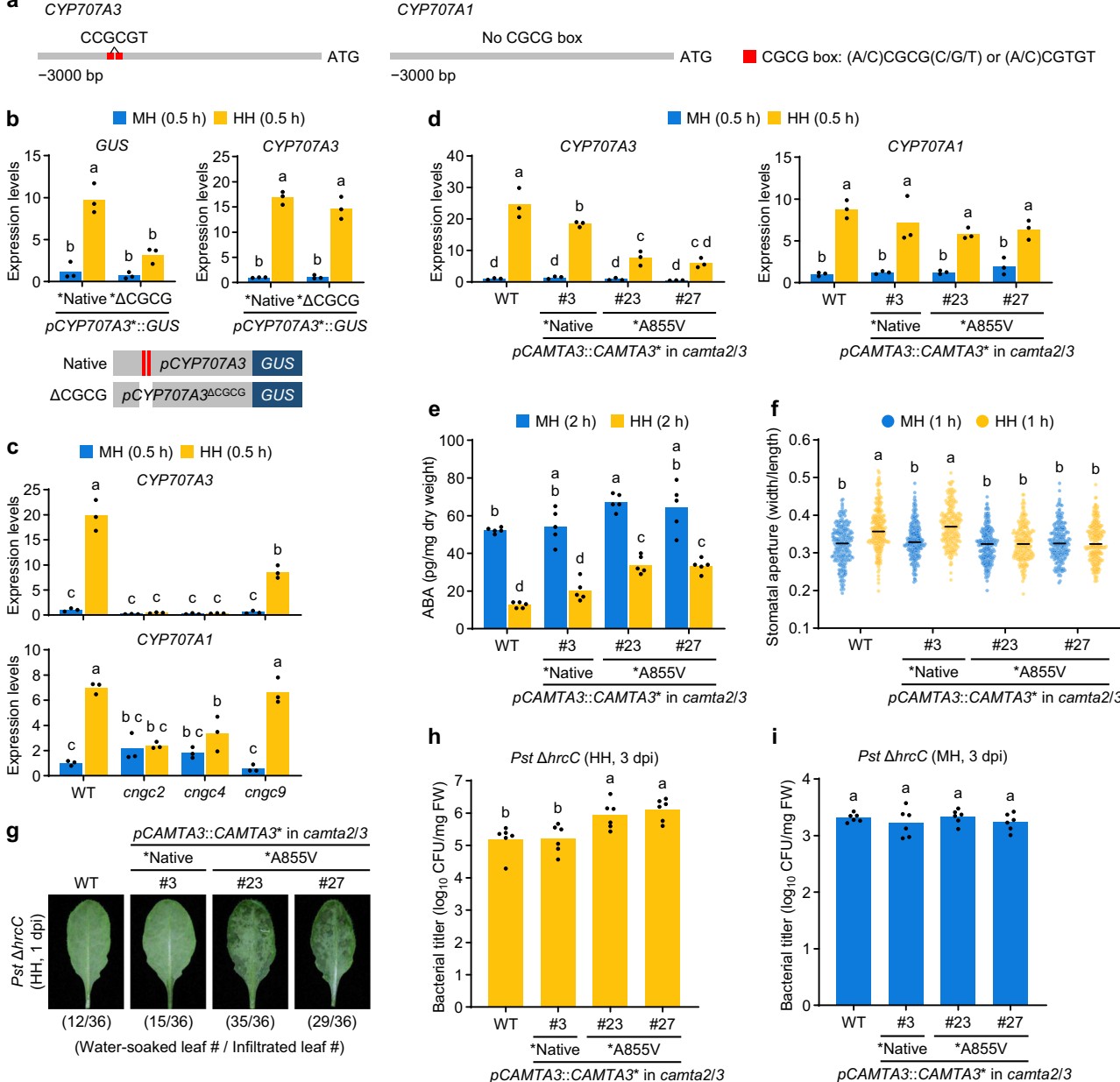

**Fig. 4 | CAMTA3 drives *CYP707A3* expression to promote resistance to bacterial water-soaking under high humidity. a** Schematic diagram of the promoter regions of *CYP707A3* and *CYP707A1*, highlighting the CGCG-box. **b** *GUS* expression driven by the *CYP707A3* promoter, with or without CGCG boxes, in humidity-treated leaves. The indicated constructs were transiently expressed in *NahG* leaves via *Agrobacterium* infiltration. Infiltrated plants were maintained under moderate humidity (MH) for 24 h, then exposed to MH or high humidity (HH) for 0.5 h. Bars represent means (*n* = 3). **c, d** Expression levels of *CYP707A3* and *CYP707A1* in humidity-treated leaves of *cngc* (**c**) and *camta3* (**d**) mutants. The indicated plants were exposed to MH or HH for 0.5 h. Bars represent means (*n* = 3). **e** ABA levels in humidity-treated leaves. The indicated plants were exposed to MH or HH for 2 h. Bars represent means (*n* = 5). **f** Stomatal aperture in humidity-treated leaves. The indicated plants were exposed to MH or HH for 1 h. Bars represent the median (WT

MH, *n* = 224; WT HH, *n* = 247; Native #3 MH, *n* = 221; Native #3 HH, *n* = 222; A855V #23 MH, *n* = 226; A855V #23 HH, *n* = 223; A855V #27 MH, *n* = 220; A855V #27 HH, *n* = 235). **g** Water-soaking in *Pst* Δ*hrcC*-inoculated leaves. The indicated plants were infiltrated with *Pst* Δ*hrcC* (OD600 = 0.02) and maintained under HH for 1 day. The number of water-soaked leaves was pooled from two independent experiments. **h, i** Bacterial growth in *Pst* Δ*hrcC*-inoculated leaves. The indicated plants were infiltrated with *Pst* Δ*hrcC* (OD600 = 0.02) and maintained under HH (**h**) or MH (**i**) for 3 days. Bars represent means (*n* = 6). Sample size (*n*) in (**b**–**e**, **h**, **i**) and (**f**) indicate biological replicates and stomata, respectively. Different letters indicate statistically significant differences (*P* < 0.05; two-way ANOVA followed by Tukey's test in (**b**–**f**); one-way ANOVA followed by Tukey's test in (**h**, **i**)). WT wild-type, CAMTA3-A855V, a partial loss-of-function variant; dpi, days post-infiltration. Experiments in (**b**–**d**, **f**, **h**, **i**) were repeated twice with similar results.

(lacking 28 T3S effectors)[37]. At 8 hpi under moderate humidity, plants were shifted to moderate or high humidity for 0.5 h before RT-qPCR analysis (Fig. 5a). At this point, bacterial populations were comparable between *Pst* DC3000 and *Pst* Δ*hrcC* (Fig. 5b), minimizing potential effects of differential bacterial growth. Under these conditions, *CYP707A1* was not induced by high humidity, even in the absence of

bacterial inoculation (Fig. 5C), likely due to desensitization from water infiltration. By contrast, *CYP707A3* was significantly induced by high humidity in mock-inoculated leaves and in those inoculated with *Pst* Δ*hrcC* or *Pst* Δ28E, but this induction was markedly suppressed by *Pst* DC3000 (Fig. 5c). CAMTA3 protein accumulation was unaffected by inoculation with the WT or mutant bacteria (Supplementary Fig. 10),

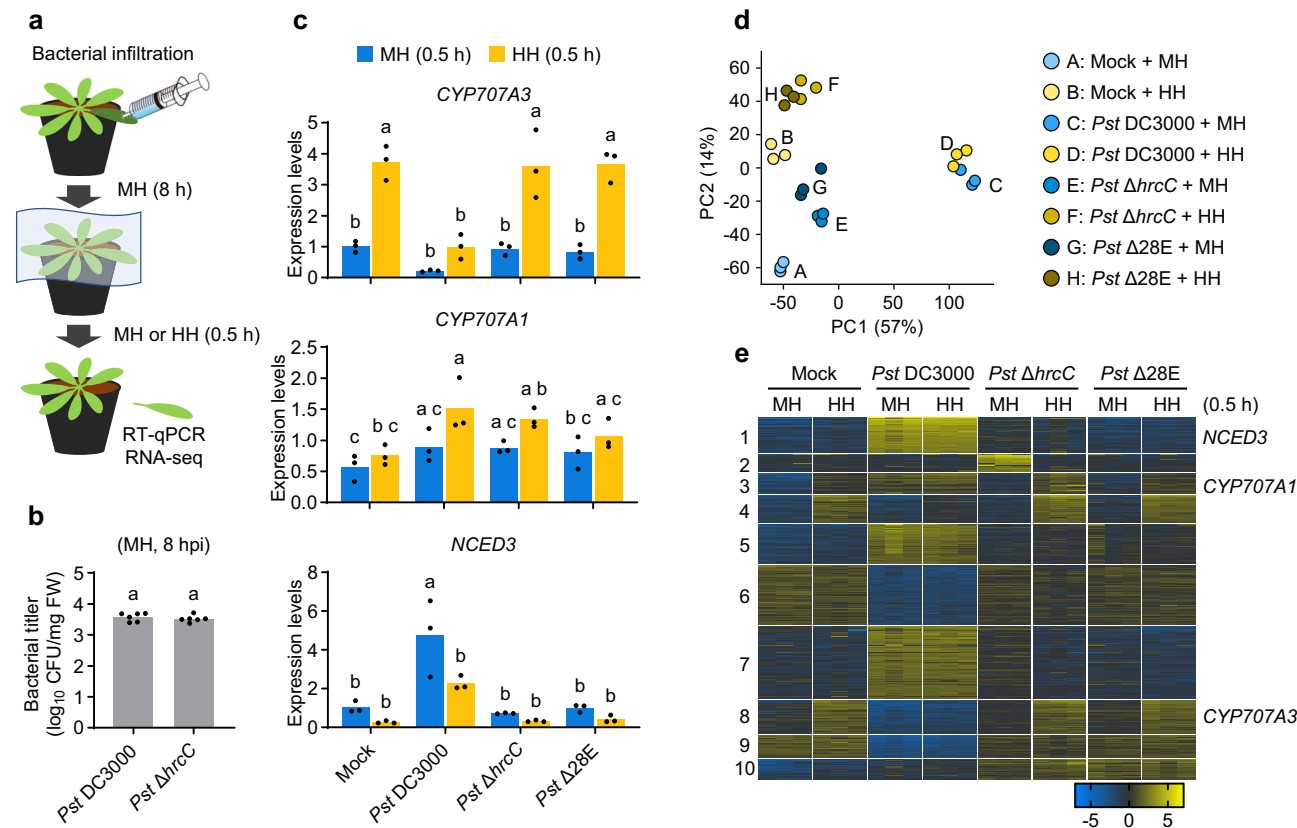

**Fig. 5 | *Pst* DC3000 blocks high humidity-driven plant transcriptome reprogramming via T3S effectors. a** Schematic overview of the experimental design for analyzing high humidity-responsive gene expression in *Pst* DC3000-inoculated leaves. Illustration of a syringe adapted from WDB Co., Ltd., Research Net (https://www.wdb.com/kenq/illust/syringe). **b** Bacterial titers in wild-type (WT) leaves infiltrated with *Pst* DC3000 or *Pst* Δ*hrcC* (OD$_{600}$ = 0.02) at 8 h post-infiltration (hpi) under moderate humidity (MH). Bars represent means ($n$ = 6). **c** Expression levels of *CYP707A3*, *CYP707A1*, and *NCED3* in *Pst* DC3000-inoculated leaves following humidity treatment. WT plants were infiltrated with water (Mock) or the indicated

*Pst* DC3000 strains (OD$_{600}$ = 0.02), maintained under MH for 8 h, and then exposed to MH or high humidity (HH) for 0.5 h. Bars represent means ($n$ = 3). **d**, **e** Transcriptome profiling of *Pst* DC3000-inoculated leaves following humidity treatment as described in (**a**, **c**). **d** Principal component analysis (PCA) of transcriptomic data. **e** K-means-clustering of the 2000 most variable genes. Sample size ($n$) indicates biological replicates. Different letters indicate statistically significant differences ($P$ < 0.05; two-tailed Welch's *t*-test in (**b**); two-way ANOVA followed by Tukey's test in (**c**)). Experiments in (**b**, **c**) were repeated twice with similar results.

suggesting that suppression is not due to CAMTA3 depletion. Consistent with previous reports[9,10], *Pst* DC3000, but not *Pst* Δ*hrcC* or *Pst* Δ28E, also induced *NINE-CIS-EPOXYCAROTENOID DIOXYGENASE 3* (*NCED3*), encoding a key ABA biosynthesis enzyme, under these conditions (Fig. 5c). Together, these results indicate that *Pst* DC3000 employs T3S effectors to suppress *CYP707A3* induction while promoting *NCED3* expression, thereby maintaining elevated ABA levels under high humidity to facilitate infection.

### *Pst* DC3000 blocks high humidity-driven plant transcriptome reprogramming via type III secretion effectors

To further investigate how *Pst* DC3000 modulates plant responses to high humidity, we performed RNA-seq analysis on leaves inoculated with *Pst* DC3000, *Pst* Δ*hrcC*, and *Pst* Δ28E for 8 h, followed by a 0.5 h of exposure to moderate or high humidity. PCA revealed a clear separation of leaf transcriptomes based on humidity conditions in non-inoculated leaves and those inoculated with *Pst* Δ*hrcC* or *Pst* Δ28E (Fig. 5d). In contrast, this separation was absent in leaves inoculated with *Pst* DC3000 (Fig. 5d), indicating that *Pst* DC3000 broadly disrupts transcriptomic responses to high humidity via T3S effectors. K-means clustering of the 2000 most variable genes resulted in 10 distinct clusters (Fig. 5e and Supplementary Data 3). High humidity-induced genes were enriched in three clusters, with Clusters 3 and 8 containing *CYP707A1* and *CYP707A3*, respectively (Fig. 5e). *Pst* DC3000 strongly suppressed high humidity-induced upregulation in Clusters 3 and 8,

and partially in Cluster 4 (Fig. 5e). Notably, Cluster 3 genes were upregulated following *Pst* DC3000 inoculation under moderate humidity, irrespective of subsequent humidity conditions (Fig. 5e). These findings indicate that *Pst* DC3000, through T3S effectors, not only suppresses CAMTA3-regulated genes but also broadly alters early transcriptional networks responsive to high humidity during infection.

### AvrPtoB contributes to suppression of *CYP707A3*-mediated water-soaking resistance

To identify the *Pst* DC3000 T3S effector(s) responsible for suppressing *CYP707A3* induction under high humidity, we analyzed multiple effector-deletion mutants of *Pst* DC3000[37]. High humidity-induced *CYP707A3* expression was significantly suppressed in WT leaves at 8 hpi with *Pst* Δ14E (lacking 14 T3S effectors) and *Pst* Δ20E (lacking 20 T3S effectors), similar to *Pst* DC3000 (Fig. 6a), at a time when T3S-dependent bacterial growth was not evident (Fig. 5b). In contrast, *Pst* Δ28E failed to suppress *CYP707A3* induction, indicating that one or more of the remaining eight effectors (HopK1, HopY1, HopB1, HopAF1, AvrPtoB, AvrPto, HopE1, and HopA1) are required for this suppression (Fig. 6a).

Given that *in-planta* expression of *AvrPtoB* has been associated with increased ABA levels through induction of *NCED3*[38], we tested whether AvrPtoB contributes to *CYP707A3* suppression. To this end, we generated transgenic plants expressing either *AvrPtoB* or its partially redundant counterpart, *AvrPto*[39], under a DEX-inducible

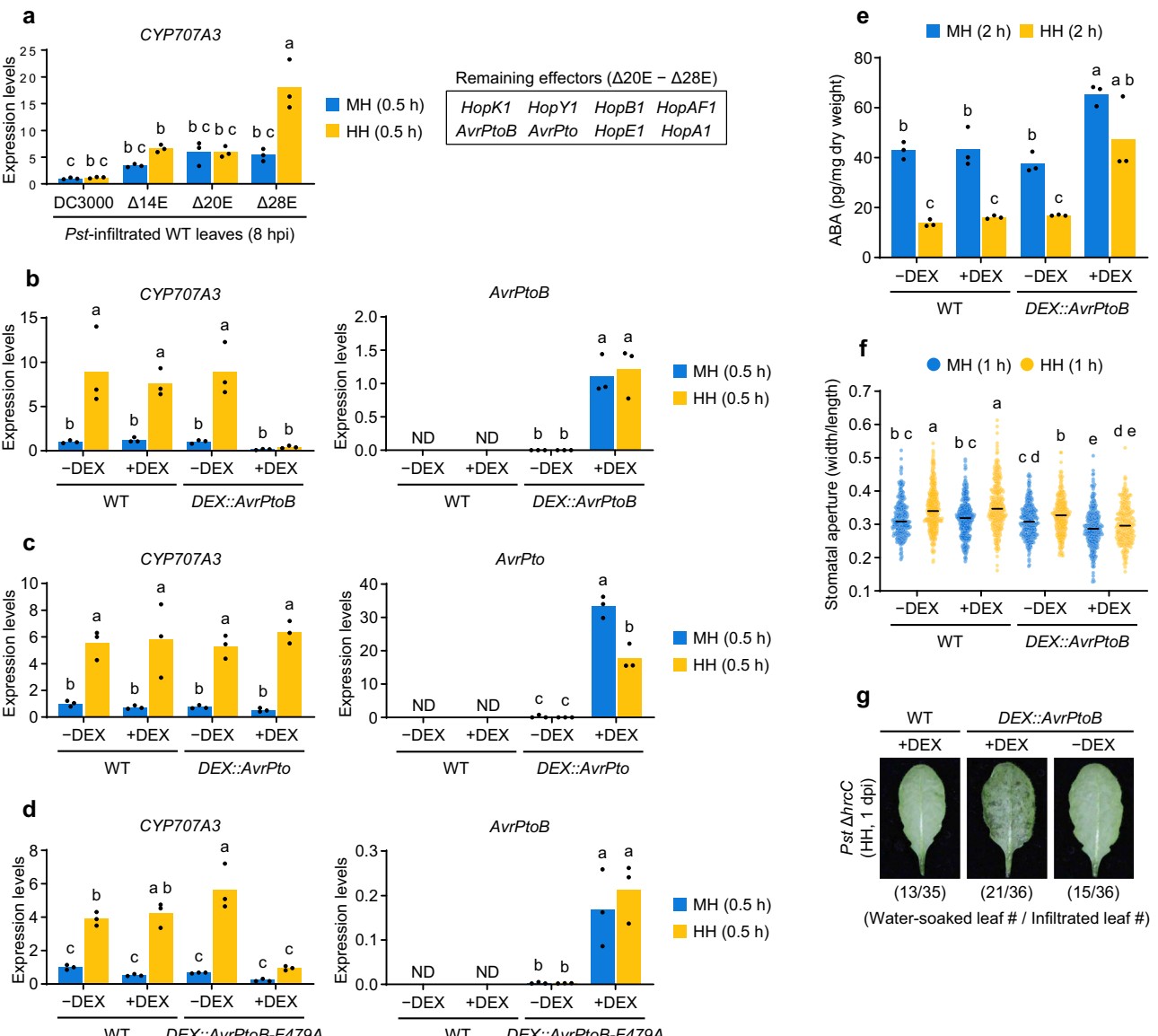

**Fig. 6 | T3S effector AvrPtoB suppresses *CYP707A3*-mediated water-soaking resistance. a** Expression levels of *CYP707A3* in *Pst* DC3000-inoculated leaves following humidity treatment. Wild-type (WT) plants were infiltrated with the indicated *Pst* DC3000 strains (OD$_{600}$ = 0.02), maintained under moderate humidity (MH) for 8 h, and then exposed to MH or high humidity (HH) for 0.5 h. Bars represent means (*n* = 3). The box highlights the T3S effectors retained in *Pst* Δ20E but absent in *Pst* Δ28E. **b**–**d** Expression levels of *CYP707A3* in humidity-treated leaves with or without induction of *AvrPtoB* (**b**), *AvrPto* (**c**), or *AvrPtoB-F479A* (**d**). The indicated plants were infiltrated with 0.1% ethanol (−DEX) or 10 μM DEX (+DEX), maintained under moderate humidity (MH) for 24 h, then exposed to MH or HH for 0.5 h. Bars represent means (*n* = 3). **e** ABA levels in humidity-treated leaves with or without *AvrPtoB* induction. The indicated plants were treated with DEX as described in (**b**–**d**), and then exposed to MH or HH for 2 h. Bars represent means (*n* = 3). **f** Stomatal aperture in humidity-treated leaves with or without

*AvrPtoB* induction. The indicated were treated with DEX as described in (**b**–**d**), and then exposed to MH or HH for 1 h. Bars represent the median (WT −DEX MH, *n* = 284; WT −DEX HH, *n* = 367; WT +DEX MH, *n* = 338; WT +DEX HH, *n* = 301; *DEX::AvrPtoB* −DEX MH, *n* = 287; *DEX::AvrPtoB* −DEX HH, *n* = 285; *DEX::AvrPtoB* +DEX MH, *n* = 284; *DEX::AvrPtoB* +DEX HH, *n* = 253). **g** Water-soaking in *Pst* Δ*hrcC*-inoculated leaves with or without *AvrPtoB* induction. The indicated plants were infiltrated with *Pst* Δ*hrcC* (OD$_{600}$ = 0.02) supplemented with −DEX or +DEX and maintained under HH for 1 day. The number of water-soaked leaves was pooled from two independent experiments. Sample size (*n*) in (**a**–**e**) and (**f**) indicate biological replicates and stomata, respectively. Different letters indicate statistically significant differences (*P* < 0.05; two-way ANOVA followed by Tukey's test). AvrPtoB-F479A, E3 ligase-deficient variant; ND, not detected. Experiments in (**a**–**d**, **f**) were repeated twice with similar results.

promoter (*DEX::AvrPtoB* and *DEX::AvrPto*; Fig. 6b, c and Supplementary Fig. 11a). Following DEX treatment, high humidity-induced *CYP707A3* expression was specifically suppressed in *DEX::AvrPtoB*, but not in *DEX::AvrPto*, despite lower protein accumulation of AvrPtoB compared to AvrPto (Fig. 6b, c and Supplementary Fig. 11b). An E3 ubiquitin ligase-deficient variant, AvrPtoB-F479A[40], retained the ability to suppress *CYP707A3* induction (Fig. 6d and Supplementary Fig. 11a), suggesting that this activity is independent of its E3 ligase function. In contrast, high humidity induction of *CYP707A1* was specifically

suppressed in *AvrPto*-expressing leaves (Supplementary Fig. 11c–e). Collectively, these findings suggest that *Pst* DC3000 selectively targets *CYP707A3* and *CYP707A1* with distinct T3S effectors, AvrPtoB and AvrPto, respectively, under high humidity.

Consistent with the predominant role for *CYP707A3* in water-soaking resistance (Fig. 1c), high humidity-triggered ABA depletion and stomatal opening were also compromised in *DEX::AvrPtoB* following DEX treatment (Fig. 6e, f). Importantly, DEX-induced *AvrPtoB* expression substantially restored water-soaking in leaves infiltrated

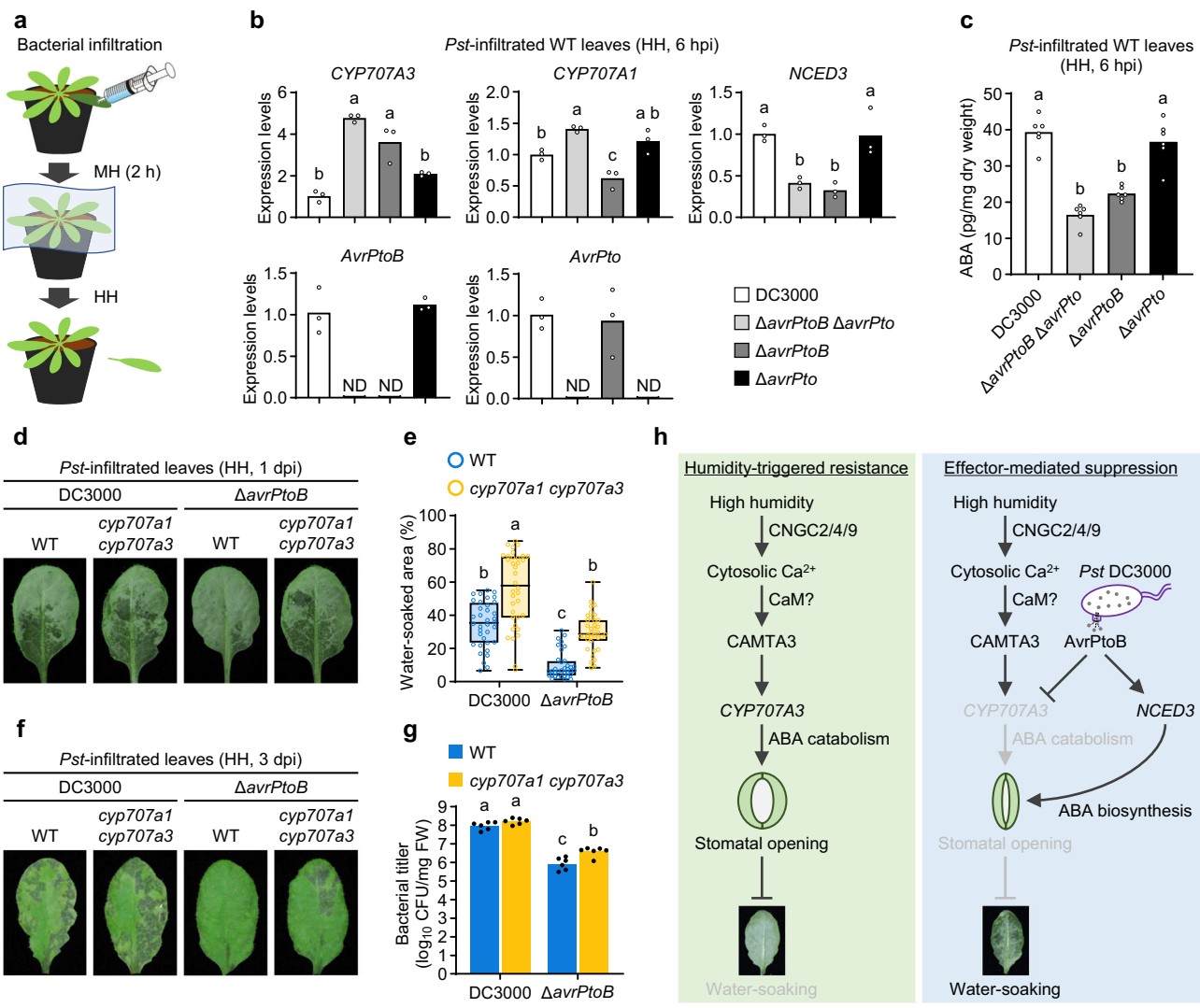

**Fig. 7 | *Pst* DC3000 employs AvrPtoB to suppress *CYP707A3* expression and promote water-soaking under high humidity. a** Schematic overview of the experimental design for analyzing gene expression, ABA accumulation, water-soaking, and bacterial growth in *Pst* DC3000-inoculated leaves. Illustration of a syringe adapted from WDB Co., Ltd., Research Net (https://www.wdb.com/kenq/illust/syringe). **b** Expression levels of *CYP707A3*, *CYP707A1*, and *NCED3* in *Pst* DC3000-inoculated leaves at 6 hours post-infiltration (hpi) under HH. Wild-type (WT) plants were infiltrated with the indicated *Pst* DC3000 strains (OD₆₀₀ = 0.2), maintained under moderate humidity (MH) for 2 h, and then transferred to HH for 4 h. Bars represent means (*n* = 3). **c** ABA levels in *Pst* DC3000-inoculated leaves. WT plants were inoculated with the indicated *Pst* DC3000 strains as described in (**b**). Bars represent means (*n* = 6). **d, e** Representative images (**d**) and quantification (**e**) of water-soaking in *Pst* DC3000-inoculated leaves. WT and *cyp707a1 cyp707a3* mutant plants were infiltrated with the indicated *Pst* DC3000 strains (OD₆₀₀ = 0.2),

maintained under MH for 2 h, and then transferred to HH for 20–22 h. Box plots represent the median, first and third quartiles, with whiskers showing the minimum and maximum values (*n* = 36; pooled from two independent experiments). **f, g** Disease symptoms (**f**) and bacterial growth (**g**) in *Pst* DC3000-inoculated leaves. WT and *cyp707a1 cyp707a3* mutant plants were infiltrated with the indicated *Pst* DC3000 strains (OD₆₀₀ = 0.0002), maintained under MH for 2 h, and then transferred to HH for 3 days. Bars represent means (*n* = 6). Sample size (*n*) in (**b**, **c**, **g**) and (**e**) indicate biological replicates and leaves, respectively. Different letters indicate statistically significant differences (*P* < 0.05; one-way ANOVA followed by Tukey's test in (**b**, **c**); two-way ANOVA followed by Tukey's test in (**e**, **g**). dpi days post-infiltration. Experiments in (**b**, **f**, **g**) were repeated twice with similar results. **h** Working model of *CYP707A3*-mediated water-soaking resistance and its suppression by *Pst* DC3000 via T3S effector AvrPtoB under high humidity.

with *Pst* DC3000 Δ*hrcC* (Fig. 6g). Together, these results indicate that AvrPtoB counteracts *CYP707A3*-mediated water-soaking resistance under high humidity, although its contribution may involve indirect effects or additional factors.

**_Pst_ DC3000 employs AvrPtoB to suppress _CYP707A3_ expression and promote water-soaking under high humidity**
We next evaluated the physiological relevance of AvrPtoB and AvrPto in promoting ABA-dependent bacterial infection under high humidity using *Pst* DC3000 mutant strain lacking both *AvrPtoB* and *AvrPto* (*Pst* Δ*avrPtoB* Δ*avrPto*). Plants were shifted from moderate to high humidity at 2 hpi, following conditions conventionally used in previous

studies[8–10] (Fig. 7a and Supplementary Fig. 12a). RT-qPCR analysis revealed that *CYP707A3* expression was suppressed from 6 to 24 hpi with *Pst* DC3000, despite continuous high humidity from 2 hpi (Supplementary Fig. 12b). This suppression was absent with *Pst* Δ28E, whereas with *Pst* Δ*avrPtoB* Δ*avrPto* it was impaired at 6 hpi but restored at 12–24 hpi (Supplementary Fig. 12b), indicating that AvrPtoB and/or AvrPto mediate early suppression of *CYP707A3*, while other effectors contribute to later suppression. Consistently, ABA levels increased at 6 hpi with *Pst* DC3000, but not with *Pst* Δ*avrPtoB* Δ*avrPto* or *Pst* Δ28E (Supplementary Fig. 12c). This initial increase was followed by further accumulation at 12 and 24 hpi with *Pst* DC3000 and partially with *Pst* Δ*avrPtoB* Δ*avrPto*, but not with *Pst* Δ28E. Together, these

findings indicate that AvrPtoB- and/or AvrPto-mediated suppression of *CYP707A3* is essential for the initial phase of ABA accumulation under high humidity, which in turn facilitates subsequent water-soaking and bacterial proliferation.

To dissect individual contributions of these effectors, we generated *Pst* DC3000 mutant strains lacking either *AvrPtoB* (*Pst ΔavrPtoB*) or *AvrPto* (*Pst ΔavrPto*) (Fig. 7b and Supplementary Fig. 13a, b). The two effectors did not affect each other's expression during infection (Fig. 7b). Consistent with DEX induction (Fig. 6b, c), high humidity-induced *CYP707A3* expression was suppressed by *Pst* DC3000 and *Pst ΔavrPto*, but permitted with *Pst ΔavrPtoB* and *Pst ΔavrPtoB ΔavrPto* (Fig. 7b). By contrast, enhanced *CYP707A1* expression relative to *Pst* DC3000 was observed only with *Pst ΔavrPtoB ΔavrPto* (Fig. 7b), suggesting redundancy in *CYP707A1* suppression. For *NCED3* induction, *Pst ΔavrPto* resembled *Pst* DC3000, whereas *Pst ΔavrPtoB* and *Pst ΔavrPtoB ΔavrPto* failed to induce it (Fig. 7b), indicating a predominant role of AvrPtoB in both *CYP707A3* suppression and *NCED3* induction. Correspondingly, ABA accumulation at 6 hpi was enhanced with *Pst* DC3000 and *Pst ΔavrPto*, but not with *Pst ΔavrPtoB* or *Pst ΔavrPtoB ΔavrPto* (Fig. 7c). Together, these results indicate that AvrPtoB drives ABA accumulation under high humidity, acting through *CYP707A3* suppression and *NCED3* induction.

We next assessed the impact of AvrPtoB on bacterial water-soaking and infection. Under high humidity, *Pst* DC3000 induced extensive water-soaking, which was strongly reduced in leaves infiltrated with *Pst ΔavrPtoB* or *Pst ΔavrPtoB ΔavrPto*, while *Pst ΔavrPto* retained partial activity (Supplementary Fig. 13c, d). These results indicate a predominant role for AvrPtoB, with a moderate contribution from AvrPto. Importantly, delayed suppression of *CYP707A3* by *Pst ΔavrPtoB ΔavrPto* at 12–24 hpi (Supplementary Fig. 12b) was insufficient to promote water-soaking, highlighting the importance of early ABA depletion for this virulence activity. Moreover, *Pst ΔavrPtoB* induced smaller water-soaked area compared to *Pst* DC3000 even in *cyp707a1 cyp707a3* leaves (Fig, 7d, e). This reduction correlated with milder disease symptoms and lower bacterial growth in both WT and *cyp707a1 cyp707a3* plants (Fig. 7f, g). These findings indicate that AvrPtoB also targets an additional, *CYP707A1/CYP707A3*-independent layer of water-soaking resistance, which may align with the additive roles of BAK1/BKK1/CERK1-mediated PTI and ABA catabolism-mediated defenses in restricting bacterial growth under high humidity (Supplementary Fig. 4d, e).

## Discussion

Terrestrial plants are constantly exposed to fluctuations in air humidity and have evolved elaborate adaptation mechanisms[17,19,41–43]. High humidity, often following prolonged rainfall, substantially exacerbates plant diseases caused by fungal and bacterial pathogens[8,44]. It is therefore plausible that plants pre-condition or activate defense responses in anticipation of these threats when sensing increased humidity. In parallel with our study, the involvement of a humidity-induced CNGC2/4-CAMTA3 module in activating *CYP707A3* expression was recently reported[22], yet its physiological relevance in plant-microbe interactions remained unresolved. Here, we demonstrate that the CNGC2/4/9-CAMTA3-CYP707A3 signaling module contributes to bacterial resistance by limiting water-soaking under high humidity (Fig. 7h). Our findings suggest that elevated humidity is perceived as a danger signal to enhance defense readiness. This aligns with earlier studies that high humidity induces ROS accumulation in Arabidopsis leaf apoplast, a hallmark of defense responses during pathogen challenge[19,21]. Additionally, high humidity increases cuticle permeability, which has been associated with enhanced resistance to the fungal pathogen *Botrytis cinerea*[18,45,46]. Rainfall and submergence have also been linked to strengthened disease resistance in plants[35,47]. These findings suggest that plants possess the mechanisms linking the sensing of increased water or humidity levels to the activation or reinforcement of defense responses. Our findings reveal a mechanism linking humidity perception to the activation of defenses that prevent bacterial access to leaf water.

Antagonistic interactions between ABA and SA signaling have been well documented[24,25,48,49], yet their modulation under varying environmental conditions remain under-explored. In Arabidopsis, *CYP707A3* overexpression enhances transpiration rates[50]. ABA application suppresses systemic acquired resistance (SAR) inducers acting both upstream and downstream of SA accumulation[24]. Enhanced SA production under continuous light suppresses *Pst* DC3000-induced water-soaking without interfering with *NCED3* induction[16]. We show that CYP707A3-mediated ABA depletion confers water-soaking resistance independently of SA under high humidity, whereas ABA deficiency enhances bacterial resistance in an SA-dependent manner under moderate humidity. These observations underscore a humidity-dependent differentiation in bacterial virulence strategies and the role of ABA depletion in bacterial resistance. The distinct regulation of SA-ABA antagonism aligns with the decrease in SA production and effectiveness under high humidity[21]. Notably, however, genes associated with SA defense, such as *SARD1*, *CBP60g*, and *BCS1*[51–53], are induced under high humidity even in the absence of pathogen challenge (Cluster 3 in Fig. 5e and Supplementary Data 3), suggesting potential coupling between humidity sensing and SA-mediated resistance, which may be further potentiated by PRR signaling (Supplementary Fig. 4e).

To our knowledge, this study provides the first evidence that phytopathogens promote infection by suppressing high humidity responses in host plants through effector proteins. Our RNA-seq analyses (Fig. 5d, e) demonstrate that *Pst* DC3000 employs multiple T3S effectors to inhibit humidity-triggered transcriptomic changes, thereby circumventing *CYP707A3*-mediated water-soaking resistance (Fig. 7h). Interestingly, *Pst* DC3000 T3S effectors also induce a subset of humidity-inducible genes, including *CYP707A1*, regardless of humidity conditions (Fig. 5e). The functional significance of their induction represents an open question for future studies. *CYP707A1* is predominantly induced in guard cells under high humidity[17], and *Pst* DC3000 elevates ABA levels in guard cells[10]. Whether bacterial *CYP707A1* induction occurs specifically in guard cells or in other tissues remains to be determined. Suppression of *CYP707A3* induction alone is sufficient to permit ABA-mediated stomatal closure, facilitating water-soaking under high humidity, as indicated by the impaired water-soaking resistance in *cyp707a3* mutants (Fig. 1c). Recent findings indicate that post-invasion stomatal reopening, rather than the initial closure to restrict bacterial entry, is critical for resisting bacterial water-soaking and infection[54], emphasizing the role of CYP707A-mediated ABA depletion as a core defense barrier limiting bacterial access to apoplastic water.

We identify AvrPtoB as a T3S effector suppressing this defense. Genetic analyses show that among the 36 T3S effectors, AvrPtoB has a role in blocking high humidity-induced *CYP707A3* expression (Fig. 7h). Earlier studies reported that AvrPtoB can target CYP707A3 and its paralogs for proteasomal degradation when these proteins are transiently expressed in *Nicotiana benthamiana*[55]. By suppressing ABA catabolism and simultaneously promoting ABA biosynthesis via *NCED3* induction[38,55], AvrPtoB manipulates both arms of host ABA balance, facilitating stomatal closure and maintaining elevated ABA levels under high humidity. These effects on *CYP707A3* and *NCED3* expression may arise indirectly as downstream consequences of AvrPtoB activity. How AvrPtoB, known to act on membrane-localized immune receptors and transcription cofactors[56–61], contributes to the modulation of *CYP707A3* and *NCED3* expression remains to be elucidated.

Our work identifies CAMTA3 transcription regulator and CNGC2/4/9 $Ca^{2+}$ channels as key regulators of humidity-induced *CYP707A3* expression, consistent with recent findings[22]. While the oomycete effector AVRblb2 has been shown to suppress CNGC activity[62], the

potential targeting of CNGCs by *Pst* DC3000 T3S effectors remains unexplored. Notably, the presence of a dominant-negative CAMTA3-A855V allele significantly impairs resistance to bacterial and fungal pathogens, SAR, as well as early transcriptional responses shared by PTI, effector-triggered immunity, and abiotic stress responses[63–65]. This underscores the critical role of CAMTA3 in promoting and/or priming defense activation under high humidity. The disruption of *CAMTA1/2/3* is also linked to nucleotide-binding leucine-rich repeat receptor activation leading to SA defense activation and autoimmunity[36,66], suggesting that CAMTAs might be targeted by pathogen effectors to counteract humidity-induced defenses. Future studies on AvrPtoB and other T3S effectors, including HopAM1, HopM1, and AvrE, which promote ABA responses[9,43,67], and their interactions with the CNGC2/4/9-CAMTA3-CYP707A3 signaling module and another water-soaking defense(s) will deepen our understanding of the interplay between humidity-triggered defenses and effector-mediated host manipulation. This work advances our knowledge of how plants and pathogens optimize immunity and virulence strategies under high humidity, driven by their competition for water as a decisive factor.

## Methods

### Plant materials and growth conditions

Arabidopsis accession Columbia-0 (Col-0) was used as the wild-type (WT) in this study. The *cyp707a1 cyp707a3 ost2-3D*, *cyp707a1 cyp707a3 sid2-2*, *aao3 sid2-2*, *bak1-5 bkk1 cerk1* (*bbc*), and *cyp707a1 cyp707a3 bbc* mutants were generated by genetic crossing. The genotypes of Arabidopsis mutants and transgenic lines were verified by PCR, Sanger sequencing, or chemical selection. Plant materials and genotyping primers used are listed in Supplementary Data 4, 6, respectively.

Surface-sterilized seeds were sown on half-strength Murashige and Skoog (MS) medium [2.3 g/L Murashige and Skoog Plant Salt Mixture (FUJIFILM Wako Pure Chemical, Cat# 392-00591), 1% (w/v) sucrose, 0.5 g/L MES, 1% (w/v) agar (FUJIFILM Wako Pure Chemical, Cat# 016-11875), pH 5.6–5.8 adjusted by KOH] followed by stratification for 1–4 days in the dark at 4°C. Plates were placed vertically in a growth chamber [22°C, 16 h light/8 h dark (long-day) or 8 h light/16 h dark (short-day), illuminated with daylight white fluorescent lamps (2 lamps/side)]. Seedlings were then transplanted into a soil mixture [50% (v/v) Super Mix A (Sakata Seed), 50% (v/v) Vermiculite G20 (Nittai)] and cultivated in either a growth chamber [22°C, 60% RH, 8 h light/16 h dark, illuminated with daylight white fluorescent lamps (2 lamps/side)] or a growth room [22°C, 30-70% RH, 16 h light/8 h dark, illuminated with daylight white fluorescent lamps (4 lamps/shelf)]. Plants were irrigated with either tap water or a modified Hoagland solution [1.67 mM $KNO_3$, 0.67 mM $MgSO_4$, 5.21 μM $KH_2PO_4$, 28.3 μM KCl, 15.5 μM $H_3BO_4$, 3.07 μM $MnCl_2$, 0.28 μM $ZnSO_4$, 0.11 μM $CuSO_4$, 13.3 μg/L 80% molybdic acid (FUJIFILM Wako Pure Chemical, Cat# 134-03332), 1.67 mM $Ca(NO_3)_2$, 46.3 μM Fe(III)-EDTA (Dojindo, Cat# 345-01245)].

For general cultivation, 2-week-old seedlings were transplanted and grown under long-day conditions with irrigation using the modified Hoagland solution. For experiments using 5-week-old plants, 2-week-old seedlings were transplanted and grown under short-day conditions with irrigation using the modified Hoagland solution. For experiments using 2-week-old plants, 4-day-old seedlings were transplanted and grown under long-day conditions with irrigation using tap water.

### Plasmid construction

For the generation of Arabidopsis transgenic plants, the promoter region (−2776 bp to −1 bp relative to the start codon) and the genomic complementation sequence (−2776 to +1930 bp relative to the start codon) of *CYP707A3* were amplified from Arabidopsis WT genomic DNA. The full-length coding sequences (excluding the stop codon) of AvrPtoB and AvrPto were amplified from *Pst* DC3000 genomic DNA. A deletion of the CGCG boxes in the *CYP707A3* promoter (−2220 bp to −2150 bp relative to the start codon) and the AvrPtoB-F479A substitution were introduced by overlap extension PCR. PCR amplifications were performed using PrimeSTAR GXL DNA Polymerase (Takara Bio, Cat# R050A). Amplified products were cloned into pENTR/D-TOPO using either the pENTR/D-TOPO Cloning Kit (Thermo Scientific, Cat# K240020SP) or the In-Fusion Snap Assembly Master Mix (Takara Bio, Cat# 638948). The resulting entry clones were verified by Sanger sequencing with the BigDye Terminator v3.1 Cycle Sequencing Kit (Applied Biosystems, Cat# 4337455) and recombined with destination vectors using Gateway LR Clonase II Enzyme mix (Invitrogen, Cat# 11791020). The following destination vectors were used: pGWB533[68] for GUS reporter constructs, pGWB501 (C-3×FLAG) for *CYP707A3* complementation constructs, and pTA70001-DEST (Addgene, Cat# 71745) for DEX-inducible constructs.

The pGWB501 (C-3×FLAG) vector was generated by cloning the 3×FLAG tag coding sequence from pAMPAT-GW-3×FLAG[69] into the AfeI site of pGWB501[68] using In-Fusion Snap Assembly Master Mix. For constructs encoding C-terminally 3×FLAG-tagged AvrPtoB, AvrPto, and AvrPtoB-F479A, entry clones were recombined with pAMPAT-GW-3×FLAG, and the resulting coding sequences were subsequently cloned into pENTR/D-TOPO.

For the generation of *Pst* DC3000 mutant strains, the flanking 1 kb regions upstream and downstream of either *avrPtoB* or *avrPto* were amplified from *Pst* DC3000 genomic DNA by PCR using PrimeSTAR GXL DNA Polymerase. Amplified fragments were assembled and cloned into EcoRI- and XbaI-digested pK18mobsacB using In-Fusion Snap Assembly Master Mix. The resulting constructs were verified by Sanger sequencing with the BigDye Terminator v3.1 Cycle Sequencing Kit. The pK18mobsacB vector was obtained from the National BioResource Project (NIG, Japan): *E. coli*.

All cloning procedures were performed according to the manufacturer's instructions. Plasmids generated in this study and the primers for cloning are listed in Supplementary Data 5, 6, respectively.

### Generation of *Arabidopsis* transgenic plants

Expression vectors were introduced into *Agrobacterium tumefaciens* strain GV3101::pMP90 by electroporation. Transgenic plants were generated by transforming WT or *cyp707a3* mutant plants using the floral dip method[70]. For the *CYP707A3* complementation experiments, T2 segregating lines were analyzed, whereas T3 or T4 homozygous lines were selected and used for all other experiments. The transgenic plants used in this study are listed in Supplementary Data 4.

### Generation of *Pst* DC3000 mutant strains

*Pst* DC3000 mutant strains lacking either *avrPtoB* or *avrPto* were generated by introducing the pK18mobsacB-based constructs into *Pst* DC3000, as previously described[71]. Deletions of *avrPtoB* and *avrPto* were verified by PCR using EmeraldAmp MAX PCR Master Mix (Takara Bio, Cat# RR320A) and Sanger sequencing with the BigDye Terminator v3.1 Cycle Sequencing Kit. The primers used for PCR and Sanger sequencing are listed in Supplementary Data 6.

### Chemicals

The following stock solutions were prepared and stored at −30°C: $LaCl_3$ (FUJIFILM Wako Pure Chemical, Cat# 123-04222), 1 M in water; $GdCl_3$ (FUJIFILM Wako Pure Chemical, Cat# 078-02661), 1 M in water; Acetosyringone (FUJIFILM Wako Pure Chemical, Cat# 320-29611), 100 mM in DMSO; Dexamethasone (DEX) (FUJIFILM Wako Pure Chemical, Cat# 047-18863), 10 mM in ethanol.

### Humidity treatment

High humidity treatment was performed by adding approximately 200 mL of modified Hoagland solution or tap water (for 2-week-old plants grown under long-day conditions) to a plastic container (Daiso,

Cat# 4550480269726), spraying tap water onto the container walls, placing the plants inside, sealing the container with plastic wrap, and immediately returning it to the growth conditions. Moderate humidity treatment was performed by adding approximately 500 mL of modified Hoagland solution or tap water (for 2-week-old plants grown under long-day conditions) to a plastic tray (As One, Cat# 1-4618-01), placing the plants inside, and immediately returning it to the growth conditions.

For inhibitor treatments and *Pst* DC3000 inoculation experiments, the indicated solutions were infiltrated into three leaves per 5-week-old WT plant using a needleless syringe (Terumo, Cat# SS-01T). Excess solution on the leaf surfaces was gently removed using tissue paper (Elleair, Cat# DSI20703128). The infiltrated plants were maintained under their growth conditions for 24 h (for inhibitor treatments) or 8 h (for *Pst* DC3000 inoculation), followed by humidity treatment. During *Pst* DC3000 inoculation experiments, lights were kept on continuously from inoculation until sampling.

### Agrobacterium-mediated transient expression
*Agrobacterium*-mediated transient expression was performed as previously described[72] with minor modifications. *Agrobacterium tumefaciens* strain GV3101::pMP90 carrying the relevant expression vectors was cultured overnight at 28°C with shaking (180 rpm) in 2×YT liquid medium [1% (w/v) yeast extract, 1.6% (w/v) tryptone, 0.5% (w/v) NaCl] supplemented with 50 µg/mL gentamicin and 100 µg/mL spectinomycin. Bacterial cells were collected by centrifugation at $2000 \times g$ for 10 min at 25°C, washed once with sterile water, and resuspended in sterile water to an $OD_{600}$ of 0.05. Acetosyringone (100 mM stock solution) was added to the suspension to a final concentration of 100 µM.

The bacterial suspensions were infiltrated into three leaves per 5-week-old *NahG* plant using a needleless syringe. Excess solution on the leaf surfaces was gently removed using tissue paper. The infiltrated plants were maintained under their growth conditions for 24 h before humidity treatment.

### *Pst* DC3000 inoculation assays
The following *Pst* DC3000 strains were used in this study: WT, Δ*hrcC*[73], Δ28E (CUCPB5585)[37], Δ14E (CUCPB5459)[74], Δ20E (CUCPB5520), Δ*avrPtoB* Δ*avrPto*[39], Δ*avrPtoB*, and Δ*avrPto*. Strains were cultured overnight at 28°C with shaking (180 rpm) in King's B liquid medium [2% (w/v) Bacto Proteose Peptone No. 3 (Gibco, Cat# 211693), 0.15% (w/v) $K_2HPO4$, 1% (v/v) glycerol, 5 mM $MgSO_4$] supplemented with 50 µg/mL rifampicin and additional antibiotics as appropriate. Bacterial cells were collected by centrifugation at $2000 \times g$ for 10 min at 25°C, washed once with sterile water, and resuspended in sterile water to the $OD_{600}$ values indicated in figure legends.

Bacterial suspensions were infiltrated into three leaves per plant using a needleless syringe. Excess solution on the leaf surfaces was gently removed using tissue paper. Infiltrated plants were maintained under their respective growth conditions for the indicated time. To ensure high humidity (~95% RH), infiltrated plants were covered with a plastic container as described in the humidity treatment.

For water-soaking assays, the abaxial sides of infiltrated leaves were photographed at 1 day post-infiltration (dpi). The number of leaves exhibiting water-soaking was recorded, or the water-soaked area relative to total leaf area was quantified using ImageJ software (https://imagej.net/ij/). For bacterial growth assays, three inoculated leaves were collected from three different plants at the indicated dpi. Leaves were homogenized in 1 mL sterile water, and 10 uL aliquots of serial dilutions ($10^{-1}$ to $10^{-8}$) were spotted onto NYGA medium [0.5% (w/v) Bacto Peptone (Gibco, Cat# 211677), 0.3% (w/v) yeast extract, 2% (v/v) glycerol. 1.5% agar] supplemented with 50 µg/mL rifampicin. Colony-forming units (CFUs) were determined after 2 days of incubation at 28°C.

### RT-qPCR
Total RNA was extracted from three leaves or a single shoot segment using Sepasol-RNA I Super G (Nacalai Tesque, Cat# 09379-55), followed by DNase treatment and reverse transcription (RT) with the PrimeScript RT reagent Kit with the gDNA Eraser (Takara Bio, Cat# RR047B), according to the manufacturer's instructions. Quantitative PCR (qPCR) was performed using Power SYBR Green PCR Master Mix (Applied Biosystems, Cat# 4368702) on a Thermal Cycler Dice Real Time System III (Takara Bio, Cat# TP950). Gene expression levels were calculated using the ΔΔCt method, with normalization to *ACTIN2* or *gap-1* as internal reference genes. Primers used for qPCR are listed in Supplementary Data 6.

### Immunoblotting
Frozen plant tissue (three leaves per sample) was ground into a fine powder and resuspended in protein extraction buffer [62.5 mM Tris-HCl (pH 6.8), 10% glycerol, 2% SDS, 5% (v/v) 2-mercaptoethanol, 0.02% bromophenol blue] supplemented with 1 mM EDTA (pH 8.0) and protease inhibitor cocktail (Roche, Cat# 11873580001). The buffer was added at a ratio of 3 µL per mg of tissue weight. Extracts were incubated at 95°C for 5 min and subsequently centrifuged at $20,000 \times g$ for 10 min at 25°C, and the supernatants were collected.

Proteins were separated on a 10% polyacrylamide gel and transferred to a PVDF membrane (Millipore, Cat# IPVH00010). The membrane was blocked 2% (w/v) skim milk (Nacalai Tesque, Cat# 31149-75) in TBS-T [25 mM Tris, 192 mM glycine, 0.1% (v/v) Tween-20] for 1 h at 25°C with agitation, rinsed once with deionized water, and once with TBS-T. The blocked membrane was incubated with the primary antibody solution overnight at 4°C with agitation, rinsed five times with deionized water, and washed once in TBS-T for 5 min at 25°C with agitation. Subsequently, the membrane was incubated with the secondary antibody solution for 1 h at 25°C with agitation, rinsed five times with deionized water, and washed once with TBS-T for 5 min at 25°C with agitation.

Chemiluminescent detection was performed using Chemi-Lumi One L (Nacalai Tesque, Cat# 07880) and a FUSION FX imaging system (Vilber). The following antibodies were used for immunoblotting: anti-FLAG (Sigma-Aldrich, Cat# F1804; 1:5000 dilution), anti-GFP (MBL, Cat# 598; 1:5000 dilution), anti-mouse IgG HRP-linked (Cell Signaling Technology, Cat# 7076; 1:5000 dilution), and anti-rabbit IgG HRP-linked (Cell Signaling Technology, Cat# 7074; 1:5000 dilution). All antibodies were diluted in TBS-T.

### Phytohormone quantification
SA, ABA, and JA-Ile were extracted and purified from freeze-dried samples as previously described[75]. Frozen samples were lyophilized for 12 h using a freeze dryer (TAITEC-OnLine; VD-250R) and pulverized with a bead-based crusher (Shake Master, Bio Medical Science; BMS-A20TP). The powder was extracted twice with 500 µL and 1.0 mL of acetonitrile/water (80:20 v/v). The extraction buffer for the first step was supplemented with internal standards, including $d_6$-SA, $d_6$-ABA, and $^{13}C_6$-JA-Ile for the quantification of SA, ABA, and JA-Ile, respectively. During each extraction step, samples were homogenized at room temperature (18–22°C) for 5 min. The samples were then centrifuged at $5900 \times g$ for 5 min at 25°C, and the supernatants from both extraction steps were combined into a 2 mL tube. Aliquots (1.1–1.2 mL) of the combined extracts were evaporated to dryness using a vacuum centrifuge (Thermo Fisher Scientific; SPD121P). The dried samples were homogenized at room temperature for 5 min and stored at −20°C for at least 30 min. After removal from the freezer, the samples were completely thawed, homogenized again for 5 min at room temperature, and centrifuged at $20,600 \times g$ for 5 min at 4°C. The resulting supernatants (50–100 µL) were filtered through a 0.22 µm filter (Corning) and centrifuged at $800 \times g$ for 3 min at 4°C. Finally, aliquots (50–80 µL) were transferred into HPLC vials (polypropylene,

12 × 32 mm screw-neck vial; Waters). Plant hormones were analyzed by an LC-MS/MS system (ACQUITY UPLC I-Class PLUS FTN/Xevo TQ-S micro; Waters) equipped with a short ODS column (TSKgel ODS-120H 2.0 mm I.D. x 5 cm, 1.9 μm; TOSHO) or a long ODS column (TSKgel ODS-120H 2.0 mm I.D. x 15 cm, 1.9 μm; TOSHO). A binary solvent system composed of 0.1% acetic acid (v/v) ACN (A) and 0.1% acetic acid (v/v) $H_2O$ (B) was used. SA, ABA, and JA-Ile were measured using the short ODS column, and separation was performed at a flow rate of 0.2 mL/min with the following linear gradient of solvent B: 0–2 min (1–5%), 2–5 min (5–15%), 5–5 min (15–40%), 25.1–27 min (100%), and 27.1–30 min (1%). The retention times for each compound were as follows: $d_6$-SA (7.39), SA (7.46), $d_6$-ABA (12.59), ABA (12.67), $^{13}C_6$-JA-Ile (21.64), and JA-Ile (21.94). The MS/MS analysis was performed under the following conditions: capillary (kV) = 1.00, source temperature (°C) = 150, desolvation temperature (°C) = 500, cone gas flow (L/h) = 50, desolvation gas flow (L/h) = 1000. Ion mode was negative. The MS/MS transitions (m/z) and collision energies (V) for each compound were as follows: $d_6$-SA (142.1/98.0, 18 V), SA (137.0/93.4, 18 V), $d_6$-ABA (269.1/159.1, 8 V), ABA (263.1/153.1, 8 V), $^{13}C_6$-JA-Ile (328.2/136.1, 20 V), and JA-Ile (322.2/130.1, 20 V). Data analysis were performed using MassLynx 4.2 (Waters).

## Stomatal aperture measurement

Following humidity treatment, detached leaves were immediately immersed in 4% (v/v) formaldehyde and incubated overnight at 25°C. Images of stomata on the abaxial leaf surface were captured using a Leica DMI6000 B microscope equipped with a 40× objective lens. Stomatal aperture (pore width/pore length) was quantified using ImageJ Fiji software (https://imagej.net/software/fiji/downloads).

## Real-time cytosolic $Ca^{2+}$ imaging

Real-time cytosolic $Ca^{2+}$ imaging of whole p35S::GCaMP3 plants was conducted as described previously[76,77]. Two-week-old plants grown under long-day conditions were used for $Ca^{2+}$ imaging. Fluorescence imaging was initiated immediately following humidity treatment. To reduce unintended stimulation by excitation light, plants were pre-exposed to light of a wavelength proximal to the excitation wavelength for ~20 min before humidity treatment.

Fluorescence signals were captured using an Axio Zoom stereo-microscope (ZEISS) equipped with an ORCA-Flash4.0 V2 digital CMOS camera (Hamamatsu) and a mercury lamp (motorized Intensilight Hg Illuminator, Nikon), with excitation at 470 nm and emission at 535 nm. Images were acquired for ~17 min with a 2 s exposure and a 5 s interval, during which the excitation light was switched off between exposures. Quantification of GCaMP3 fluorescence was performed using ZEN Pro (ZEISS). Fluorescence intensity across the leaf surface was measured, and relative fluorescence values were calculated by normalizing each time point to the initial intensity (set to 1).

## RNA-sequencing and data analysis

Total RNA was extracted from three leaves and treated with DNase using the NucleoSpin RNA Plant kit (Macherey-Nagel, Cat# 740949.50) according to the manufacturer's instructions. RNA quality was assessed using a bioanalyzer (Agilent Technologies).

Each raw read data stored in DDBJ with accession numbers DRR545835 to DRR545846 was generated and analyzed as follows. Each RNA library was prepared from 1000 ng of total RNA using the NEBNext Ultra II RNA Library Prep Kit for Illumina (New England Biolabs, Cat# E7770) according to the manufacturer's instructions. PCR for library amplification was performed for seven cycles. The average length of the library was 350 bp, as measured by a bioanalyzer. Concentrations were measured by Kapa Library Quantification Kit (Kapa Biosystems, Cat# KK4824), each sample was diluted to 10 nM, and equal volumes were mixed. Single reads of 75 bp were obtained using a NextSeq 500 system (Illumina). CLC genomics workbench version 21

(Qiagen) was used to trim raw reads and map to the TAIR10 Arabidopsis genome (https://www.arabidopsis.org/). In trimming, the following parameters were changed (quality limit = 0.001, number of 5′ terminal nucleotides =15, number of 3′ terminal nucleotides = 5, and minimum number of nucleotides in reads = 25) and other options were default values. In mapping, all options were default values.

The generation and analysis of each raw read data stored in DDBJ with accession numbers DRR545847 to DRR545870 differ from the above analysis in three points. (1) Each library was prepared from 500 ng total RNA using the NEBNext Ultra II Directional RNA Library Prep Kit for Illumina (New England Biolabs, Cat# 7760). PCR for library amplification was performed for eight cycles. Each sample was diluted to 4 nM, and equal volumes were mixed. (2) Single reads of 100 bp were obtained using a NextSeq 1000 system (Illumina). (3) CLC genomics workbench version 24 (Qiagen) was used for read trimming and mapping. In trimming and mapping, the following parameters were changed (minimum number of nucleotides in reads = 40, and strand setting = Revers). All other methods, numbers, and options are the same as the above analysis.

The obtained read counts were processed and analyzed using iDEP 2.01 (https://bioinformatics.sdstate.edu/idep20/) with default settings. Gene Ontology and cis-regulatory enrichment analyses were performed using ShinyGO 0.81 (https://bioinformatics.sdstate.edu/go/) with default settings. For cis-regulatory enrichment analysis, the AGRIS database was used as the pathway reference.

## Accession numbers

The gene sequences analyzed in this study were obtained from The Arabidopsis Information Resource (https://www.arabidopsis.org/) and The Pseudomonas Genome Database (https://www.pseudomonas.com/), under the accession numbers listed in Supplementary Data 7.

## Statistical analysis

All data, except RNA-seq, were analyzed using GraphPad Prism 10 software (https://www.graphpad.com/features). No statistical methods were used to predetermine sample size, and no data were excluded from analyses. Experiments were not randomized, and investigators were not blinded to group allocation or outcome assessment. Statistical comparisons were performed using two-tailed Welch's t-test, or one-way or two-way ANOVA followed by Tukey's multiple comparison test, as specified in the figure legends.

## Reporting summary

Further information on research design is available in the Nature Portfolio Reporting Summary linked to this article.

## Data availability

The raw RNA-sequencing data have been deposited in the DDJB database under accession code PRJDB17982. Source data are provided with this paper.

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

## Acknowledgements

We thank Chika Tateda, Mie Matsubara, Taiga Ishihara, Natsuki Tsuchida, and Temma Takazawa (Nara Institute of Science and Technology) for technical assistance; Yuri Kanno (RIKEN CSRS) for supporting plant hormone analysis; Akira Mine (Kyoto University) for *cyp707a* seeds; Atsushi Takemiya (Yamaguchi University) for *ost2-3D* seeds; Michael F. Thomashow (Michigan State University) for *camta* seeds; Alan Collmer (Cornell University) for *Pst* DC3000 strains. We also thank the NAIST Life Science Collaboration Center (LiSCo) for supporting this study. This work was supported in part by JSPS KAKENHI Grant Numbers 21K14829 (S.Y.), 24H00565 (M.T.) and 21H02507 (Y.S.), JST ACT-X Grant Number JPMJAX22BN (S.Y.), ERATO Grant Number JPMJER2403 (M.T.), Cooperative Research Grant of the Genome Research for BioResource, NODAI Genome Research Center, Tokyo University of Agriculture (S.Y.), and Institute for Fermentation, Osaka Grant Number G-2025-2-089 (S.Y.).

## Author contributions

S.Y. and Y.S. conceived the study. S.Y. designed the research, conducted most of the experiments, and analyzed the data. A.R., T.H., and S.U. performed part of the RT-qPCR and *Pst* DC3000 inoculation assay. Y.W., Y.T., and M.O. performed phytohormone quantification. H.I., R.Suzuki, and M.T. performed GCaMP3 imaging. A.S., R.Sk, and I.Y. performed RNA-sequencing. S.Y. and Y.S. wrote the manuscript with input from the other authors.

## Competing interests

The authors declare no competing interests.
