## [Transparent Peer Review file · Nature Communications]

Humidity-driven ABA depletion determines plant-pathogen competition for leaf water

Corresponding Author: Dr Shigetaka Yasuda

Version 0:

Reviewer comments:

Reviewer #1

(Remarks to the Author)

This study explores the mechanisms underlying plant resistance against bacterial pathogens under high humidity conditions, focusing on the impact of elevated humidity on ABA-mediated defense responses. The authors demonstrate that high humidity induces ABA depletion through the expression of CYP707A3, which is driven by Ca²⁺ signaling via calcium channels CNGC2/4/9 and transcriptional activation by the regulator CAMTA3. This ABA depletion promotes stomatal opening, thereby limiting bacterial water-soaking. Furthermore, the *Pseudomonas syringae* effector AvrPtoB counteracts this defense mechanism by suppressing CYP707A3 induction and promoting ABA biosynthesis, thereby facilitating pathogen colonization. The study attempts to link humidity-driven plant water dynamics, ABA level and pathogen effector functions, and provides new insights into humidity-influenced plant defense and anti-defense.

My main concerns:

1. I noticed many of the main conclusions of the study, including high humidity induction of CYP707A3, the involvement of CNGCs in high humidity-induced calcium flux, the role of CAMTA3 in the transcriptional induction of CYP707A3 gene, were reported in a recent publication (Hussain et al., PNAS 2024). While the authors connect CYP707A3 induction and stomatal closure process to water soaking and pathogen resistance, the role of ABA and stomata in water soaking and disease resistance was reported in several previous studies as well, which diminishes the novelty of these findings.

2. The authors demonstrated that Pst DC3000 effector AvrPtoB acts to suppress CYP707A3 expression; however, this suppression was observed in transgenic plants overexpressing avrPtoB, but not the avrPtoB-deleted Pst DC3000 strain. This raises concerns about potential artifact effects from overexpressing the effector. Does the avrPtoB-deleted strain exhibit differences in ABA level or water soaking compared to Pst DC3000 strain? In addition, prior studies have suggested that AvrPtoB targets the plant PRR complex and is supposed to localize to the membrane of the plant cell. Therefore, the AvrPtoB's suppression of CYP707A3 is likely indirect? What is the potential molecular mechanism of CYP707A3 suppression by AvrPtoB?

Other comment:

Figure 2a-b, the differences in stomatal aperture between MH and HH, as well as between WT and cyp707a1a3 plants, seem fairly small.

Reviewer #2

(Remarks to the Author)

In this work, the authors reported the discovery of a CNGC2/4/9-CAMTA3-CYP707A3 signaling cascade in the regulation of ABA levels of plants under high humidity conditions. They also uncovered an inhibitory effect of the avirulent effector AvrPtoB on CYP707A3 induction, which facilitates leaf water-soaking and bacterial colonization upon inoculation. This work is interesting and proposes a model for CYP707A3-mediated resistance to foliar phytopathogens. However, I have a few suggestions that I believe will improve the manuscript.

Here are the four major concerns towards this manuscript:

1. The authors stated that the avirulent effector AvrPtoB facilitates leaf water-soaking under high humidity conditions by

suppressing the expression level of CYP707A3. However, they did not show evidence demonstrating significant differences in the water-soaking phenotype and bacterial titers between wild-type leaves inoculated with Pst DC3000 and those infiltrated with Pst avrPto avrPtoB.

2. One of the most interesting findings in this paper is the CNGC2/4/9-CAMTA3-CYP707A3 module in the regulation of ABA catabolism upon HH treatment. However, there is a lack of genetic evidence to substantiate this conclusion. I recommend that the authors use additional methods (like ChIP-qPCR) to confirm the change in binding affinity of CAMTA3 to the CYP707A3 promoter in the presence and absence of CNGC2/4/9. Furthermore, they need to perform epistasis analyses using appropriate genetic materials to reinforce their claim.

3. It is known that the induction of CYP707A1/A3 in wild-type leaves upon HH treatment only sustains for a couple of hours (see Okamoto et al. 2009 Plant Physiology), whereas the water-soaking appearance of leaves won't be evident until 24 h post inoculation. I am not surprised to see an exaggerating water-soaking phenotype in cypa1a3 due to the constitutively increased ABA levels in this mutant. However, to build an intimate association between CYP707A3 transcriptional regulation and the resistance to water-soaked leaves, I recommend the authors examine time-resolved expression patterns of CYP707A3 in wild-type plants inoculated with Pst DC3000, Pst hrcC, and Pst avrPto avrPtoB. It is important to note that the period of sampling for RT-qPCR experiments should match the time allowing appearance of leaf water-soaking.

4. The authors suggested that high humidity-induced cytosolic Ca²⁺ elevation precedes CYP707A3 induction, and the CNGC2/4/9 channels and the Ca²⁺ signature decoder CAMTA3 play predominant roles here. However, there is no experimental evidence to compare the increasing kinetics of cytosolic Ca²⁺ and CYP707A3 transcripts. Also, I recommend they perform the same experiment in Fig. S5 to compare cytosolic Ca²⁺ dynamics in WT, cngc2/4/9 and camta3 plants upon MH/HH treatments.

Minor issues:

1. The results of water-soaking phenotype characterization are inconsistent throughout the whole paper (e.g. WT leaves inoculated with Pst hrcC at 1 dpi under HH), and some of the images of mutant leaves in supplementary figures look fake. The authors should refer to the well-established methods published previously (e.g. Xin et al. Nature 2016 Vol 539 and Hu et al. Cell Host & Microbe 2022 Vol 30), and increase the quality of leaf imaging. Moreover, it is recommended that they use Log₁₀ CFU/cm² when plotting the data of bacterial growth curve experiments, as the disruption of ABA catabolism might affect the density of fresh leaves.

2. Since the authors claimed that the regulation of ABA-catabolizing enzyme CYP707A3 plays an important role in leaf water-soaking, it is necessary for them to measure the dynamic changes in the levels of major metabolites (e.g. phaseic acid/PA and dihydrophaseic acid/DPA) of the major ABA catabolic pathway. To further strengthen the manuscript, I suggest they measure time-resolved changes in PA and DPA levels in wild-type leaves infiltrated with Pst DC3000 and Pst avrPto avrPtoB. Please refer to the paper published by Okamoto et al. (2009) in Plant Physiology for more technical details.

3. The differences in stomatal aperture across different plant genotypes under MH and HH conditions are very small. Therefore, I recommend the authors examine transpiration rate, a physiological parameter that is more relevant to the ratio of leaf water loss.

4. The overall language and logical flow of the manuscript should be further revised to improve clarity.

Reviewer #3

(Remarks to the Author)

In this study, the authors investigated the role of CYP707A1/A3 in *Pseudomonas syringae* pv. tomato (Pst)-induced water soaking in *Arabidopsis* and dissected the signaling pathway upstream of CYP707A1/A3. High - humidity-induced calcium signaling, which is controlled by CNGC2 and CNGC4, activates the transcription factor CAMTA3. In turn, CAMTA3 induces the activation of CYP707A3, thereby suppressing stomatal closure and rescuing water soaking. Pst utilizes AvrPtoB to suppress CYP707A3 expression and trigger water soaking for more efficient infection in plants. Overall, the manuscript is highly readable, the evidence is clear, and the conclusion is well founded. However, there is still one minor point that needs to be addressed as follows:

The authors demonstrated the CYP707A3 expression level, ABA content, and stomatal aperture in DEX::AvrPtoB plants to illustrate the role of AvrPtoB in the suppression of stomatal behavior via CYP707A3. But what about the water soaking phenotype in WT and DEX::AvrPtoB plants with and without DEX treatment, when induced by Pst hrcC under moderate humidity (MH) and high humidity (HH) conditions?

Version 1:

Reviewer comments:

Reviewer #1

(Remarks to the Author)

My comments remain similar to the last round. Although the authors provided evidence for the role of bacteria-delivered AvrPtoB in CYP707A3 expression, the mechanism is still unknown. Based on previous understandings of AvrPtoB, this effect is likely indirect.

Reviewer #3

(Remarks to the Author)

The authors clearly addressed my concerns. I do not have additional questions on the manuscript.

Reviewer #1 (Remarks to the Author):

This study explores the mechanisms underlying plant resistance against bacterial pathogens under high humidity conditions, focusing on the impact of elevated humidity on ABA-mediated defense responses. The authors demonstrate that high humidity induces ABA depletion through the expression of *CYP707A3*, which is driven by Ca^{2+} signaling via calcium channels CNGC2/4/9 and transcriptional activation by the regulator CAMTA3. This ABA depletion promotes stomatal opening, thereby limiting bacterial water-soaking. Furthermore, the *Pseudomonas syringae* effector AvrPtoB counteracts this defense mechanism by suppressing *CYP707A3* induction and promoting ABA biosynthesis, thereby facilitating pathogen colonization. The study attempts to link humidity-driven plant water dynamics, ABA level and pathogen effector functions, and provides new insights into humidity-influenced plant defense and anti-defense.

Response:

Thank you very much for positive evaluation of our work.

My main concerns:

1. I noticed many of the main conclusions of the study, including high humidity induction of *CYP707A3*, the involvement of CNGCs in high humidity-induced calcium flux, the role of CAMTA3 in the transcriptional induction of *CYP707A3* gene, were reported in a recent publication (Hussain et al., PNAS 2024). While the authors connect *CYP707A3* induction and stomatal closure process to water soaking and pathogen resistance, the role of ABA and stomata in water soaking and disease resistance was reported in several previous studies as well, which diminishes the novelty of these findings.

Response:

We appreciate the reviewer's insightful comment. While the CNGC2/4–CAMTA3–*CYP707A3* module was indeed recently implicated in humidity responses (Hussain et al., PNAS 2024), its role in host–microbe interactions has not been addressed. Our study reveals that this module plays a crucial role in leaf water defense, enhancing plant resistance to bacterial water-soaking and infection under high humidity. Specifically, we demonstrate that this defense mechanism is induced in response to increased humidity and counteracts ABA-mediated stomatal closure—a key virulence strategy employed by foliar bacterial pathogens. Moreover, we demonstrate that this defense response is

specifically suppressed by the bacterial effector AvrPtoB, underscoring its functional relevance as a barrier to bacterial infection. We believe our findings advance current knowledge by providing mechanistic insight into how plants integrate environmental cues with immune responses and how bacteria counteract the host defenses, thereby illuminating an underexplored aspect of plant–microbe–environment interactions that is of both fundamental and applied significance.

2. The authors demonstrated that *Pst* DC3000 effector AvrPtoB acts to suppress *CYP707A3* expression; however, this suppression was observed in transgenic plants overexpressing *avrPtoB*, but not the *avrPtoB*-deleted *Pst* DC3000 strain. This raises concerns about potential artifact effects from overexpressing the effector. Does the *avrPtoB*-deleted strain exhibit differences in ABA level or water soaking compared to *Pst* DC3000 strain?

Response:

We thank the reviewer for raising this important point. In the revised manuscript, we have included new data quantifying ABA levels and water-soaking in wild-type leaves inoculated with a *Pst* DC3000 mutant strain lacking AvrPtoB (*Pst* Δ *avrPtoB*). Compared with the wild-type strain, *Pst* Δ *avrPtoB* induced significantly lower ABA accumulation and markedly reduced water-soaking (Fig. 7c–e). These data support our conclusion that AvrPtoB suppresses *CYP707A3*-mediated resistance and mitigate concerns regarding potential artifacts arising from effector overexpression in transgenic plants. Collectively, the results further substantiate AvrPtoB as a key virulence factor that modulates host ABA catabolism during infection.

In addition, prior studies have suggested that AvrPtoB targets the plant PRR complex and is supposed to localize to the membrane of the plant cell. Therefore, the AvrPtoB's suppression of *CYP707A3* is likely indirect? What is the potential molecular mechanism of *CYP707A3* suppression by AvrPtoB?

Response:

We appreciate the reviewer's thoughtful comment. AvrPtoB is indeed known to function as an E3 ubiquitin ligase that targets membrane-localized PRR components such as FLS2, BAK1, and CERK1 for degradation. However, our data show that *CYP707A3*-mediated water soaking resistance remains intact even when these canonical AvrPtoB targets are

simultaneously disrupted (Supplementary Fig. 4d), suggesting that suppression of *CYP707A3* occurs independently of these PRR pathways.

To further address this, we examined an E3 ligase-deficient variant of AvrPtoB and found that it still suppresses humidity-induced *CYP707A3* expression (Fig. 6d). Moreover, infection with *Pst* DC3000 did not alter CAMTA3 protein accumulation (Supplementary Fig. 10), indicating that AvrPtoB does not simply act through destabilization of this key transcriptional regulator. These results collectively point to a previously unsuspected mechanism of AvrPtoB action, independent of both its E3 ligase activity and established targets.

While our findings provide initial evidence for such a novel mechanism, we recognize that they do not yet resolve the underlying molecular basis. Elucidating this mechanism will require a substantial volume of work beyond the scope of the current study, but we believe our results provide an important foundation and an intriguing new direction for future investigation.

Other comment:

Figure 2a-b, the differences in stomatal aperture between MH and HH, as well as between WT and *cyp707a1a3* plants, seem fairly small.

Response:

We thank the reviewer for this observation. To address this point, we increased the number of biological replicates for the stomatal aperture measurements, which strengthened the statistical support for our conclusions. Further details are provided in our response to Reviewer #2, minor concern 3 (Please see below).

Reviewer #2 (Remarks to the Author):

In this work, the authors reported the discovery of a CNGC2/4/9-CAMTA3-CYP707A3 signaling cascade in the regulation of ABA levels of plants under high humidity conditions. They also uncovered an inhibitory effect of the avirulent effector AvrPtoB on CYP707A3 induction, which facilitates leaf water-soaking and bacterial colonization upon inoculation. This work is interesting and proposes a model for CYP707A3-mediated resistance to foliar phytopathogens. However, I have a few suggestions that I believe will improve the manuscript.

Response:

Thank you very much for your positive evaluation of our work.

Here are the four major concerns towards this manuscript:

1. The authors stated that the avirulent effector AvrPtoB facilitates leaf water-soaking under high humidity conditions by suppressing the expression level of *CYP707A3*. However, they did not show evidence demonstrating significant differences in the water-soaking phenotype and bacterial titers between wild-type leaves inoculated with *Pst* DC3000 and those infiltrated with *Pst* $\Delta avrPto$ $\Delta avrPtoB$.

Response:

We thank the reviewer for raising this important comment. As noted in our response to Reviewer 1, in the revised manuscript, we have included direct comparisons of water-soaking phenotypes and bacterial titers between wild-type plants inoculated with *Pst* DC3000 and those inoculated with a $\Delta avrPtoB$ mutant strain (Fig. 7d-g). These data show that bacterial water-soaking and growth under high humidity are significantly reduced by deletion of *AvrPtoB*, supporting our hypothesis that AvrPtoB contributes to suppression of *CYP707A3* expression and thereby promotes water-soaking during infection.

2. One of the most interesting findings in this paper is the CNGC2/4/9-CAMTA3-CYP707A3 module in the regulation of ABA catabolism upon HH treatment. However, there is a lack of genetic evidence to substantiate this conclusion. I recommend that the authors use additional methods (like ChIP-qPCR) to confirm the change in binding affinity of CAMTA3 to the *CYP707A3* promoter in the presence and absence of

CNGC2/4/9. Furthermore, they need to perform epistasis analyses using appropriate genetic materials to reinforce their claim.

Response:

We appreciate the reviewer's thoughtful comment and agree that additional experimentation—such as ChIP-qPCR for CAMTA3 occupancy and genetic epistasis studies—would further strengthen the significance of the CNGC2/4/9–CAMTA3–CYP707A3 module. In the current study, we provide multiple convergent lines of evidence consistent with our model: (i) rapid high humidity (HH)-triggered cytosolic Ca²⁺ elevation (GCaMP3) that precedes *CYP707A3* induction (Supplementary Fig. 7b, d); (ii) pharmacological Ca²⁺ channel blockade suppressing *CYP707A3* but not *CYP707A1* (Supplementary Fig. 7c); (iii) impaired HH induction of *CYP707A3* in *cngc2* and *cngc4* with a decrease in *cngc9* (Fig. 4c and Supplementary Fig 8a, b); (iv) loss of HH inducibility with CAMTA3-A855V (Fig. 4d); and (v) functional requirement for the CAMTA-binding (CGCG box) motif in the *CYP707A3* promoter (Supplementary Fig. 6d, e). Together, these data support the proposed connectivity and align with prior reports implicating the same module in humidity responses (Hussain et al., PNAS 2024).

We agree that, strictly speaking, ChIP-qPCR and epistasis analysis would provide formal proof. However, it is not yet clear whether Ca²⁺/CaM-dependent activation of CAMTA3 leads to substantial changes in DNA-binding affinity. If not, ChIP-qPCR may only detect subtle effects and would require extensive time-course assays and optimization, in addition to generating CAMTA3-tagged lines in the relevant mutant backgrounds for comprehensive epistasis studies. While we have begun preparing the necessary materials, these experiments demand substantial time and efforts and therefore extend beyond the scope of the present manuscript.

Taken together with our new and existing data, we believe the evidence strongly supports the HH-Ca²⁺ (CNGC2/4/9)-CAMTA3-CYP707A3 axis, while recognizing that definitive mechanistic dissection will be an important direction for future studies.

3. It is known that the induction of *CYP707A1/A3* in wild-type leaves upon HH treatment only sustains for a couple of hours (see Okamoto et al. 2009 Plant Physiology), whereas the water-soaking appearance of leaves won't be evident until 24 h post inoculation. I am not surprised to see an exaggerating water-soaking phenotype in *cyp707a1 cyp707a3* due to the constitutively increased ABA levels in this mutant. However, to build an intimate

association between *CYP707A3* transcriptional regulation and the resistance to water-soaked leaves, I recommend the authors examine time-resolved expression patterns of *CYP707A3* in wild-type plants inoculated with *Pst* DC3000, *Pst* Δ *hrcC*, and *Pst* Δ *avrPto* Δ *avrPtoB*. It is important to note that the period of sampling for RT-qPCR experiments should match the time allowing appearance of leaf water-soaking.

Response:

We thank the reviewer for this valuable suggestion. In the revised Supplementary Fig. 12b, we now provide time-resolved RT-qPCR data comparing *CYP707A3* expression in wild-type plants inoculated with *Pst* DC3000, Δ 28E, and Δ *avrPtoB* Δ *avrPto* strains. The results show that *CYP707A3* expression, apparent at 2 h post-infiltration (hpi), is more sustained when inoculated with the effector-deficient strains (up to 6 hpi with Δ *avrPtoB* Δ *avrPto* and up to 12-24 hpi with Δ 28E), compared to wild-type *Pst* DC3000. These results indicate that *Pst* DC3000 actively suppresses *CYP707A3* expression via T3S effectors, with AvrPtoB and/or AvrPto primarily mediating early suppression and additional effectors contributing to later attenuation. Notably, in these experiments, plants were maintained under moderate humidity for 2 h after bacterial infiltration, allowing water evaporation and disappearance of the initial water-soaked area, before being shifted to high humidity. Under these conditions, water-soaking developed at 24 hpi with *Pst* DC3000, but was dramatically reduced with *Pst* Δ *avrPtoB* Δ *avrPto* or *Pst* Δ *avrPtoB*, and was only partially observed with *Pst* Δ *avrPto* (Supplementary Fig. 13c, d). This suggests that delayed suppression of *CYP707A3* with *Pst* Δ *avrPtoB* Δ *avrPto* is insufficient to trigger water-soaking.

Together, our new data indicate that early suppression of *CYP707A3* at around 6 hpi, predominantly by AvrPtoB, is critical for the development of water-soaking at 24 hpi (1 dpi). We believe this new dataset strengthens the mechanistic link between bacterial modulation of *CYP707A3* transcriptional regulation and water-soaking susceptibility.

4. The authors suggested that high humidity-induced cytosolic Ca^{2+} elevation precedes *CYP707A3* induction, and the CNGC2/4/9 channels and the Ca^{2+} signature decoder CAMTA3 play predominant roles here. However, there is no experimental evidence to compare the increasing kinetics of cytosolic Ca^{2+} and *CYP707A3* transcripts. Also, I recommend they perform the same experiment in Fig. S5 to compare cytosolic Ca^{2+} dynamics in WT, *cngc2/4/9* and *camta3* plants upon MH/HH treatments.

Response:

We thank the reviewer for this valuable comment. In the revised manuscript, we provide additional time-course data of *CYP707A3* induction following high humidity exposure (Supplementary Fig. 7d), which indicate that cytosolic Ca^{2+} elevation precedes *CYP707A3* induction. Regarding humidity-induced cytosolic Ca^{2+} dynamics, its elevation through *CNGC2/4* has been described in Hussain et al., PNAS 2024. However, comprehensive comparisons in *cngc2/4/9* and *camta3* mutants will require the generation of new genetic materials, as noted earlier. We therefore propose to address these aspects, including the potential role of CAMTA3 in the feedback regulation of cytosolic Ca^{2+} dynamics, in future studies.

Minor issues:

1. The results of water-soaking phenotype characterization are inconsistent throughout the whole paper (e.g. WT leaves inoculated with *Pst* Δ *hrcC* at 1 dpi under HH), and some of the images of mutant leaves in supplementary figures look fake. The authors should refer to the well-established methods published previously (e.g. Xin et al. Nature 2016 Vol 539 and Hu et al. Cell Host & Microbe 2022 Vol 30), and increase the quality of leaf imaging. Moreover, it is recommended that they use Log_{10} CFU/cm² when plotting the data of bacterial growth curve experiments, as the disruption of ABA catabolism might affect the density of fresh leaves.

Response:

We appreciate the comments. Regarding the water-soaking analyses, we used the recommended experimental conditions (see response to comment 3) to obtain additional data supporting the critical role of AvrPtoB (with a moderate contribution from AvrPto) in the early suppression of *CYP707A3* expression and water-soaking. We also provide raw leaf images in the corresponding source data file, which demonstrate the consistency of water-soaking phenotypes in the WT and mutant leaves. Regarding bacterial growth quantification, we confirmed that leaf fresh weight was not significantly affected even in *cyp707a1 cyp707a3* mutants under our conditions (Fig. R1). This validates the use of leaf fresh weight for normalizing bacterial titers in our study.

Fig. R1: Growth phenotype and leaf fresh weight of *cyp707a1 cyp707a3* mutant.

a, b, Representative images (a) and leaf fresh weight (b) of 5-week-old plants grown under short-day conditions. Graph bars indicate the median ($n = 36$). Statistically significant differences between wild-type (WT) and *cyp707a1 cyp707a3* were determined by Welch's *t*-test ($P < 0.05$).

2. Since the authors claimed that the regulation of ABA-catabolizing enzyme CYP707A3 plays an important role in leaf water-soaking, it is necessary for them to measure the dynamic changes in the levels of major metabolites (e.g. phaseic acid/PA and dihydrophaseic acid/DPA) of the major ABA catabolic pathway. To further strengthen the manuscript, I suggest they measure time-resolved changes in PA and DPA levels in wild-type leaves infiltrated with *Pst* DC3000 and *Pst* Δ *avrPto* Δ *avrPtoB*. Please refer to the paper published by Okamoto et al. (2009) in *Plant Physiology* for more technical details.

Response:

We thank the reviewer for this constructive suggestion. Following the recommended approach, we measured time-course changes in ABA (Supplementary Fig. 12c), PA and DPA levels (Fig. R2) in leaves infiltrated with *Pst* DC3000, *Pst* Δ *avrPto* Δ *avrPtoB* and *Pst* Δ 28E. PA levels were reduced at 6 hpi with both the wild-type and mutant strains, but restored at 12 hpi and further increased at 24 hpi with *Pst* DC3000, while only partially with *Pst* Δ *avrPto* Δ *avrPtoB* and not with *Pst* Δ 28E. By contrast, DPA levels gradually declined throughout the time course with all strains. The results indicate that *AvrPtoB*-mediated suppression of *CYP707A3* does not necessarily produce the predicted downstream changes in canonical ABA catabolites. This suggests additional complexity in ABA catabolism and inactivation mechanisms during bacterial infection. We consider the detailed dissection of these mechanisms, including the identity and biological relevance of ABA-derived catabolites, as an important direction for future studies.

Fig. R2: PA and DPA levels in *Pst* DC3000-inoculated leaves.

a, Schematic overview of the experimental setup for analyzing phaseic acid (PA) and dihydrophaseic acid (DPA) levels. **b**, PA and DPA levels in *Pst* DC3000-inoculated leaves. Wild-type (WT) plants were infiltrated with the indicated *Pst* DC3000 strains (OD₆₀₀ = 0.2), maintained under moderate humidity (MH) for 2 h (2 hpi), and then exposed to high humidity (HH) for 4 h (6 hpi), 10 h (12 hpi), or 22 h (24 hpi). Bars indicate means ($n = 4$). Different letters indicate statistically significant differences ($P < 0.05$; two-way ANOVA followed by Tukey's test). $\Delta avrPtoB \Delta avrPto$, *Pst* DC3000 mutant lacking both *avrPtoB* and *avrPto*; $\Delta 28E$, *Pst* DC3000 mutant lacking 28 type III secretion effectors; hpi, hours post-infiltration.

3. The differences in stomatal aperture across different plant genotypes under MH and HH conditions are very small. Therefore, I recommend the authors examine transpiration rate, a physiological parameter that is more relevant to the ratio of leaf water loss.

Response:

We thank the reviewer for this insightful suggestion. We agree that the differences in stomatal aperture across plants genotypes under moderate and high humidity appear relatively small. To address this, we carefully re-examined the reproducibility of all stomatal aperture data and confirmed consistent results across independent experiments (Fig. R3a-c). We suspect that the small differences reflect the generally high humidity conditions in our experimental environment in Japan.

We fully acknowledge that transpiration rate would be an important physiological parameter for assessing leaf water loss. However, such measurements are beyond the

scope of the present study. Importantly, we believe that the reproducible stomatal aperture data, together with our complementary physiological evidence, sufficiently supports our overall conclusions.

Fig. R3: Stomatal aperture in humidity-treated leaves.

a, b, Plants were exposed to moderate humidity (MH) or high humidity (HH) for 1 h. **c,** Plants were infiltrated with 0.1% ethanol (-DEX) or 10 μ M DEX (+DEX) and maintained under MH for 24 h before humidity treatment. Bars represent the median ($n = 149-245$ in **a-exp 1**; $n = 149-264$ in **a-exp 2**; $n = 220-247$ in **b-exp 1**; $n = 154-339$ in **b-exp 2**; $n = 284-367$ in **c-exp 3**; $n = 179-361$ in **c-exp 3**). Different letters indicate statistically significant differences ($P < 0.05$; two-way ANOVA followed by Tukey's test). WT, wild-type; CAMTA3-A855V, a partial loss-of-function variant.

4. The overall language and logical flow of the manuscript should be further revised to improve clarity.

Response:

We thank the reviewer for this valuable suggestion. In the revised manuscript, we have carefully edited the overall language and improved the logical flow throughout the text to

enhance clarity and readability. We believe these revisions have substantially improved the overall presentation of our study.

Reviewer #3 (Remarks to the Author):

In this study, the authors investigated the role of CYP707A1/A3 in *Pseudomonas syringae* pv. *tomato* (*Pst*)-induced water soaking in Arabidopsis and dissected the signaling pathway upstream of CYP707A1/A3. High-humidity-induced calcium signaling, which is controlled by CNGC2 and CNGC4, activates the transcription factor CAMTA3. In turn, CAMTA3 induces the activation of *CYP707A3*, thereby suppressing stomatal closure and rescuing water soaking. *Pst* utilizes AvrPtoB to suppress *CYP707A3* expression and trigger water soaking for more efficient infection in plants. Overall, the manuscript is highly readable, the evidence is clear, and the conclusion is well founded. However, there is still one minor point that needs to be addressed as follows:

Response:

Thank you very much for positive evaluation of our work.

The authors demonstrated the *CYP707A3* expression level, ABA content, and stomatal aperture in *DEX::AvrPtoB* plants to illustrate the role of AvrPtoB in the suppression of stomatal behavior via CYP707A3. But what about the water soaking phenotype in WT and *DEX::AvrPtoB* plants with and without DEX treatment, when induced by *Pst* Δ *hrcC* under moderate humidity (MH) and high humidity (HH) conditions?

Response:

We thank the reviewer for this important question. We performed additional experiments to evaluate the water-soaking phenotype in *DEX::AvrPtoB* plants following *Pst* Δ *hrcC* infiltration under high humidity. The number of water-soaked leaves was increased in *DEX::AvrPtoB* (+DEX) compared with WT (+DEX) and *DEX::AvrPtoB* (-DEX) (Fig. 6g), further supporting the role of AvrPtoB in promoting water-soaking under high humidity. Consistently, water-soaking was substantially reduced in leaves inoculated with *Pst* DC3000 mutant strain lacking *AvrPtoB* (*Pst* Δ *avrPtoB*) compared with the WT strain (Fig. 7d, e and Supplementary Fig. 13c, d), reinforcing the conclusion that AvrPtoB is a major effector responsible for this phenotype.

Reviewer #1 (Remarks to the Author):

My comments remain similar to the last round. Although the authors provided evidence for the role of bacteria-delivered AvrPtoB in CYP707A3 expression, the mechanism is still unknown. Based on previous understandings of AvrPtoB, this effect is likely indirect.

Response:

We agree with Reviewer #1 that the mechanism remains unresolved, and that *AvrPtoB* may influence *CYP707A3* expression indirectly. In the revised manuscript (L389–391 and 510–511), we have added statements explicitly acknowledging that AvrPtoB-dependent effects on *CYP707A3* may arise as secondary or downstream consequences of its activities.

Reviewer #3 (Remarks to the Author):

The authors clearly addressed my concerns. I do not have additional questions on the manuscript.

Response:

We thank the reviewer for the positive evaluation and are pleased that our revisions fully addressed your previous concerns.